# IFIT1 is rapidly evolving and exhibits disparate antiviral activities across 11 mammalian orders

**Matthew B McDougal, Ian N Boys, Anthony M De Maria, Emi Nakahara, John W Schoggins***

Department of Microbiology, University of Texas Southwestern Medical Center, Dallas, United States

## eLife Assessment

This **important** report describes the changing antiviral activity of IFIT1 across mammals and in response to distinct viruses, likely as a result of past arms races. One of the main strengths of the manuscript is the breadth of mammalian IFIT1 orthologs and viruses that were tested, as well as the thoroughness of the positive selection analysis. Overall the evidence is **convincing**, and the discussion conveys well the limitations due to physical interactions with other IFITs that are not accounted for.

*For correspondence:
John.Schoggins@
UTSouthwestern.edu

**Abstract** Mammalian mRNAs possess an N7-methylguanosine (m7G) cap and 2'O methylation of the initiating nucleotide at their 5' end, whereas certain viral RNAs lack these characteristic features. The human antiviral restriction factor IFIT1 recognizes and binds to specific viral RNAs that lack the 5' features of host mRNAs, resulting in targeted suppression of viral RNA translation. This interaction imposes significant host-driven evolutionary pressures on viruses, and many viruses have evolved mechanisms to evade the antiviral action of human IFIT1. However, little is known about the virus-driven pressures that may have shaped the antiviral activity of IFIT1 genes across mammals. Here, we take an evolution-guided approach to show that the IFIT1 gene is rapidly evolving in multiple mammalian clades, with positive selection acting upon several residues in distinct regions of the protein. In functional assays with 39 IFIT1s spanning diverse mammals, we demonstrate that IFIT1 exhibits a range of antiviral phenotypes, with many orthologs lacking antiviral activity against viruses that are strongly suppressed by other IFIT1s. We further show that IFIT1s from human and a bat, the black flying fox, inhibit Venezuelan equine encephalitis virus (VEEV) and strongly bind to Cap0 RNAs. Unexpectedly, chimpanzee IFIT1, which differs from human IFIT1 by only eight amino acids, does not inhibit VEEV infection and exhibits minimal Cap0 RNA-binding. In mutagenesis studies, we determine that amino acids 364 and 366, the latter of which is rapidly evolving, are sufficient to confer the differential anti-VEEV activity between human and chimpanzee IFIT1. These data suggest that virus-host genetic conflicts have influenced the antiviral specificity of IFIT1 across diverse mammalian orders.

## Introduction

Interferon-induced protein with tetratricopeptide repeats 1 (IFIT1) is an interferon-stimulated gene (ISG) effector that distinguishes self and non-self RNA. Mammalian mRNAs possess an m7G cap and 2'O methylation of the initiating nucleotide, whereas many viral RNAs lack one or both features. IFIT1 binds uncapped 5'-triphosphate RNA (5'-ppp-RNA) and capped but unmethylated RNA (Cap0, an m7G

cap lacking 2'-O methylation) (*Hyde et al., 2014*; *Pichlmair et al., 2011*; *Abbas et al., 2017*). IFIT1 binding to viral RNA inhibits infection by decreasing the amount of accessible RNA from the pool of replicating viral RNA (*Diamond and Farzan, 2013*) and by interacting with eIF proteins required for translation of viral proteins (*Hyde and Diamond, 2015*).

This restriction imposes a host-driven evolutionary pressure for viruses to evade IFIT1 through various mechanisms (*Hyde and Diamond, 2015*). Some encode their own capping and 2'O methyltransferase enzymes, translate through internal ribosome entry sites (IRES), hijack host mRNA caps by cap-snatching, or mask the 5' end with RNA structures (*Hyde and Diamond, 2015*). For the alphavirus Venezuelan equine encephalitis virus (VEEV), which has a positive-sense, single-stranded RNA genome, a 5' UTR structure in the pathogenic Trinidad strain occludes IFIT1 binding, whereas a single-nucleotide change in the attenuated TC-83 strain destabilizes the structure and allows inhibition (*Hyde et al., 2014*). VEEV TC-83 is sensitive to human IFIT1 and mouse Ifit1B, indicating at least partial conservation of antiviral function by IFIT proteins. Viral countermeasures may, therefore drive adaptive changes in IFIT1, yet the scope of this pressure across mammalian IFIT1 remains unclear.

Pathogenic viruses can cause severe disease or death in mammals, imposing a major selective pressure on the host to evolve antiviral defenses. Viruses face reciprocal pressure to evade these defenses. These 'molecular arms races' leave signatures of positive selection in many immune genes. Positive selection, or rapid evolution, is denoted by a high ratio of nonsynonymous to synonymous substitutions (dN/dS >1). Computational evolutionary analyses of antiviral ISGs have revealed specific domains and residues that are rapidly evolving (*Daugherty and Malik, 2012*; *Duggal and Emerman, 2012*; *Sironi et al., 2015*). Several methods can determine if a gene, or part of a gene, is undergoing positive selection. These include Phylogenetic Analysis using Maximum Likelihood (PAML), Fast Unconstrained Bayesian AppRoximation (FUBAR), and Mixed Effects Model of Evolution (MEME). Each one is a codon-substitution model that identifies sites within a gene that evolve under selection. For example, PAML tests whether a neutral-or-purifying model (M7) fits the data significantly worse than one that includes positive selection (M8). Such approaches have pinpointed certain domains and amino acids that determine antiviral function in effectors such as MX1 and TRIM5α (*Mitchell et al., 2012*; *Sawyer et al., 2005*). In addition to evolutionary pressure from host-pathogen molecular arms races, IFIT1 evolution has also been influenced by gene loss, gene duplication, and recombination with IFIT1B (*Daugherty et al., 2016*). However, the antiviral consequences of such events have only been tested functionally in a few species, and the breadth of IFIT1 diversity across Mammalia remains unknown (*Daugherty et al., 2016*). Recently, in a screen for antiviral ISGs from bats, we found that IFIT1 from the black flying fox displays particularly strong suppression of VEEV (*Cruz-Rivera et al., 2024*). From these prior studies, we hypothesized that evolutionary pressures may have shaped 'species-specific' antiviral activity among IFIT1 orthologs. Accordingly, we sought to use evolutionary and functional methods to interrogate differences in antiviral activity and biochemical properties of mammalian IFIT1.

## Results
### IFIT1 is rapidly evolving in major clades of mammals

Many antiviral effectors exhibit signatures of positive selection (*Daugherty and Malik, 2012*; *Duggal and Emerman, 2012*), prompting us to test whether IFIT1 is similarly evolving across several diverse clades of mammals. Using PAML (*Yang, 2007*), we determined that IFIT1 exhibits signatures of rapid evolution in primates, bats, carnivores, and ungulates, with the weakest signal in carnivores (*Figure 1A*). Rodents were excluded because many species lack an IFIT1 ortholog (*Daugherty et al., 2016*). While all lineages showed signs of rapid evolution, carnivore IFIT1s exhibited the least significant likelihood of pervasive positive selection (*Figure 1A*). Furthermore, each lineage contained at least two codons under selection. To uncover other rapidly evolving sites, we applied two codon-based models that complement PAML. FUBAR uses a Bayesian framework that assumes selection as pervasive across all branches of the phylogenetic tree (*Murrell et al., 2013*). It detected at least one positively selected codon in every lineage except carnivores (*Figure 1B*). MEME identifies codons that experience episodic selection on a subset of branches (*Murrell et al., 2012*) it identified several such sites in each lineage we tested (*Figure 1C*). These data indicate that positive selection has acted on multiple regions of IFIT1 across Mammalia.

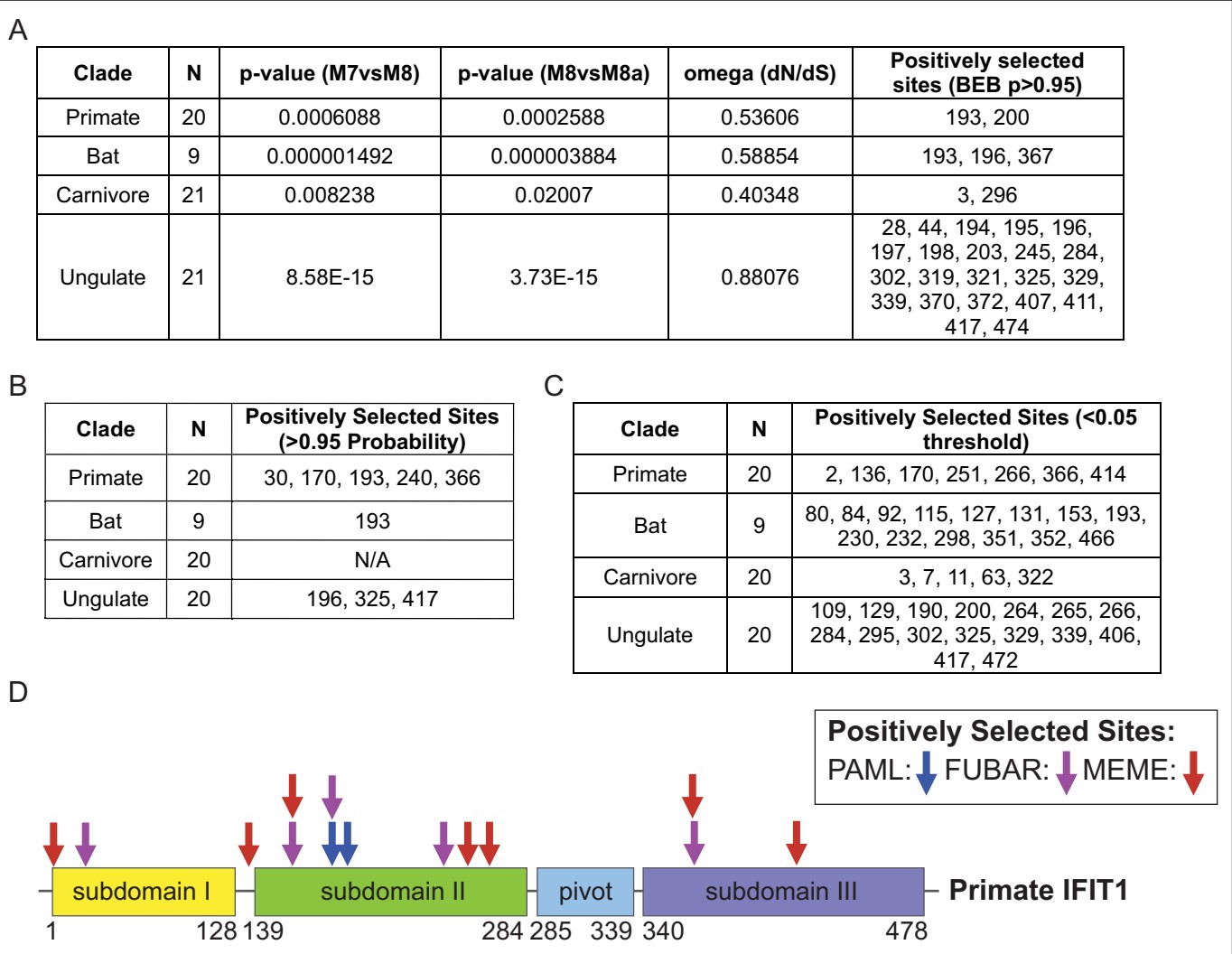

**Figure 1.** Evolutionary analysis of Interferon-induced protein with tetratricopeptide repeats 1 (IFIT1) from clades of mammals reveals rapid evolution. (**A**) Phylogenetic Analysis using Maximum Likelihood (PAML) analysis on NCBI-derived coding sequences that were aligned and then analyzed using CodeML. Likelihood ratio tests were performed to compare model 7 vs model 8 and model 8 vs model 8 a to determine the presence of positive selection. N=number of sequences input, and numbers under 'positively selected sites' represent residue number of the reference sequence. (**B**) Fast Unconstrained Bayesian AppRoximation (FUBAR) analysis on aligned coding sequences using HyPhy software and the DataMonkey application. N=number of sequences input, and numbers under 'positively selected sites' represent residue number of the reference sequence. (**C**) Mixed Effects Model of Evolution (MEME) analysis on aligned coding sequences using HyPhy software and the DataMonkey application. N=number of sequences input, and numbers under 'sites of episodic positive selection' represent residue number of the reference sequence. (**D**) Diagram of Primate IFIT1 domains and location of positively selected sites determined by PAML, FUBAR, and MEME.

The online version of this article includes the following figure supplement(s) for figure 1:

**Figure supplement 1.** Analysis of Interferon-induced protein with tetratricopeptide repeats 1 (IFIT1) for recombination and positive selection.

Recombination between paralogous genes can affect analyses of positive selection (**Kosakovsky Pond et al., 2006**), and the evolution of some IFIT1 sequences has been influenced by recombination with IFIT1B (**Daugherty et al., 2016**). To confirm that the sequences examined were likely IFIT1 orthologs, we aligned all IFIT1 sequences from this study with representative IFIT1B and IFIT3 sequences from several mammals and built a maximum-likelihood tree from their C-terminal regions, as previously described (**Daugherty et al., 2016**). All IFIT1 clustered together, separate from IFIT1B and IFIT3 sequences (**Figure 1—figure supplement 1A**), supporting their classification as orthologs. We further evaluated IFIT1 sequences for recombination breakpoints using GARD, a genetic-algorithm approach for recombination detection (**Kosakovsky Pond et al., 2006**). A single breakpoint was detected in

primates (codon 38) and ungulates (codon 188), whereas two were identified in bats (codons 33, 381) and carnivores (codons 14, 370) (*Figure 1—figure supplement 1B*).

The presence of GARD breakpoints prompted reanalysis of IFIT1 using only coding regions located after the breakpoint in primates and ungulates or between breakpoints in bats and carnivores. PAML still detected positive selection in every clade except carnivores (*Figure 1—figure supplement 1C*), which had the lowest signal of positive selection in our original analysis. Furthermore, nearly all codons flagged in the initial analysis (*Figure 1A–C*) were recovered when we re-ran FUBAR (*Figure 1—figure supplement 1D*) and MEME (*Figure 1—figure supplement 1E*) on alignments trimmed after or between the GARD breakpoints. Considering that sequences were trimmed, which removed certain sites previously identified as rapidly evolving, almost all other possible positively selected sites were validated in primates (11/12), bats (14/16), ungulates (30/37), and carnivores (2/4) across PAML, FUBAR, and MEME analyses (*Figure 1—figure supplement 1*). To further clarify whether recombination could have impacted these analyses, we used Recombination Detection Program 4 (RDP4) (*Martin et al., 2015*) to remove inferred recombinant sequences from the primate alignment. Subsequent PAML, FUBAR, and MEME analyses recovered 10 of the original 14 selected sites (*Figure 1—figure supplement 1F*). Importantly, codons 170, 193, and 366 in primates continued to exhibit evidence of rapid evolution in two independent methods in both the GARD breakpoint-trimmed and RDP4-filtered datasets. In bats, codon 193 likewise showed positive selection in two analyses of the between-breakpoint alignment. While signatures of selection differed slightly between the original, the post- or between-breakpoint trimmed, and RDP4-filtered datasets, overall trends and in many cases residue-specific signals were consistent, highlighting the robustness of the evolutionary signatures.

Overall, these analyses indicate that IFIT1 is rapidly evolving across multiple mammalian clades. Such widespread rapid evolution may result in functional consequences on antiviral activity for an individual species or more broadly across clades of mammals. To address the first prediction, we next tested for functional implications of within-species rapid evolution of IFIT1 using human IFIT1 as a model.

## Human IFIT1 residue 193 exhibits mutational resiliency

We focused on three codons in primate IFIT1 – encoding V170, L193, and L366 in the human ortholog – that were rapidly evolving in at least two models (*Figure 1D*). None of these residues have been associated with RNA-binding or antiviral function (*Pichlmair et al., 2011*; *Abbas et al., 2017*). In the crystal structure of human IFIT1 bound to RNA, V170 and L366 lie outside the RNA-binding tunnel, whereas L193 resides in the TPR4 loop. This loop sits between α-helices 9 and 10, forms a lid over the 3′ exit of the tunnel, and helps link subdomain II to subdomain III (*Figure 2A–C*; *Abbas et al., 2017*). L193 in the TPR4 loop protrudes into the tunnel exit, comes into close contact with RNA (*Figure 2C*), and is adjacent to R187, a residue essential for RNA binding and antiviral function (*Pichlmair et al., 2011*). To test the functional importance of L193, we knocked out endogenous IFIT1 in Huh7.5 cells with CRISPR–Cas9 editing and re-expressed CRISPR-resistant IFIT1 variants carrying every possible amino acid substitution at position 193. Cells were then infected with VEEV (TC-83 strain) encoding a GFP reporter (VEEV-GFP), and infection was quantified by flow cytometry (*Figure 2D*). Western blotting determined that all IFIT1 point mutants were expressed to comparable levels (*Figure 2D*, Bottom). All hydrophobic and positively charged mutants at position 193 inhibited VEEV similarly to the wild-type (193 L) IFIT1 (*Figure 2D*). Polar amino acids introduced at position 193 also retained antiviral function (*Figure 2D*). Severe loss of function in anti-VEEV activity of IFIT1 was only notable with glycine or proline (~50% loss of antiviral function) or with negatively charged residues (~75–100% loss of antiviral function) (*Figure 2D*). Thus, codon 193 displays notable functional resiliency, as most substitutions preserve antiviral activity. This result connects our computational analysis to a concrete functional outcome within one species. We next asked whether positive selection also influences interspecies differences in IFIT1 antiviral activity across diverse mammalian orthologs.

## IFIT1 orthologs from diverse mammals demonstrate species-specific antiviral activity

We next cloned 39 IFIT1 orthologs—derived from 38 mammalian species, including two predicted equine sequences—into lentiviral expression vectors. These proteins share 60–98% identity with

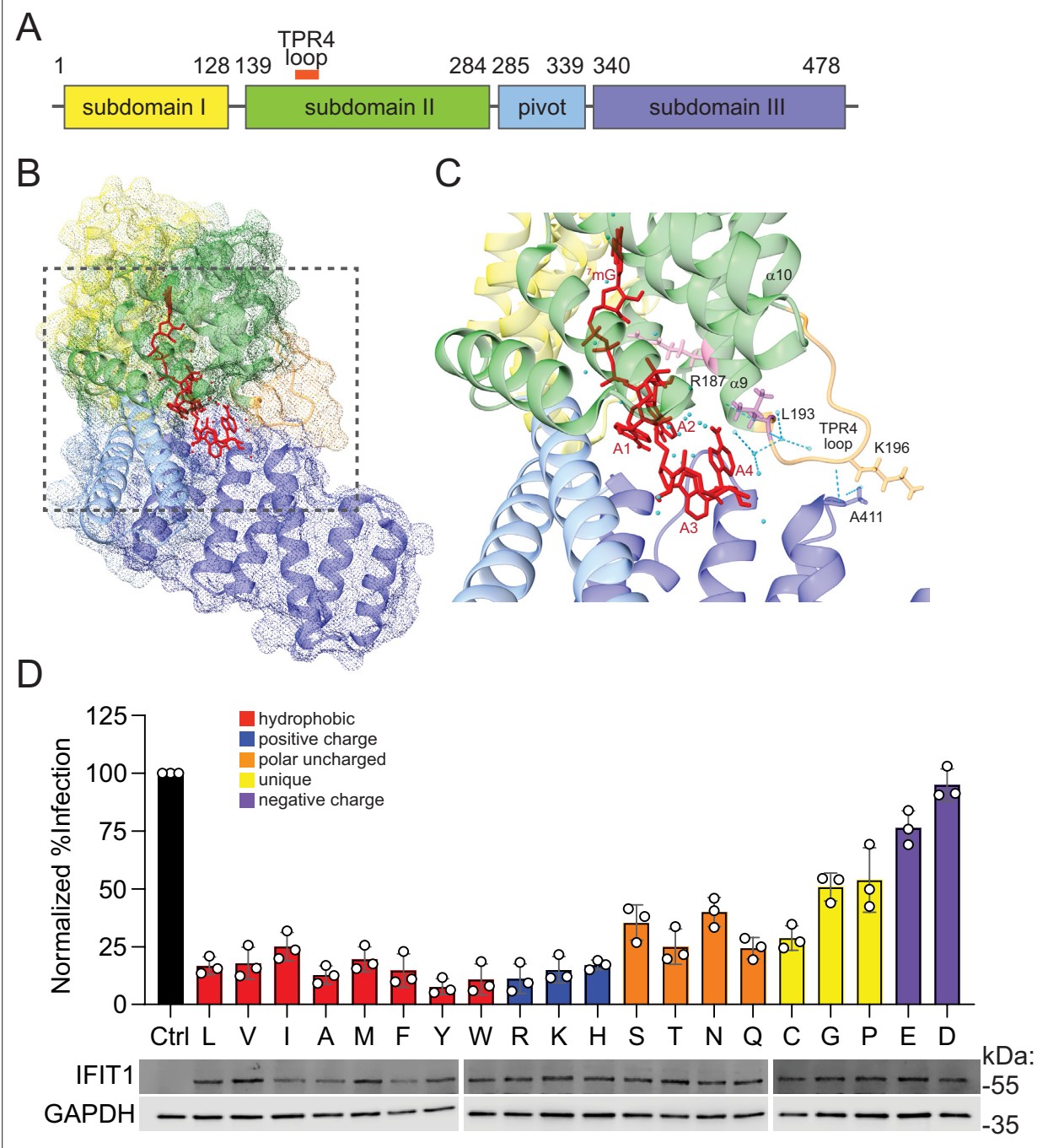

**Figure 2.** Mutagenesis of rapidly evolving residue 193 in human Interferon-induced protein with tetratricopeptide repeats 1 (IFIT1) reveals mutational resiliency. (**A**) Depiction of the human IFIT1 protein with the TPR4 loop location (orange) shown. (**B**) Solved crystal structure of IFIT1 bound to RNA with TPR4 loop forming a 'lid' over the exit of the RNA-binding tunnel (PDB:5udj). (**C**) Zoom in on the TPR4 loop and RNA-binding tunnel exit illustrating the location of the TPR4 loop and residue 193 in relation to bound RNA in the human IFIT1 crystal structure. (**D**) (Top) Saturating mutagenesis screen in cells expressing human IFIT1 with residue 193 mutated to every possible residue and challenged with Venezuelan equine encephalitis virus (VEEV) infection. Data represent mean +/- SD, n=3 independent experiments. (Bottom) Western blot from lysates of Huh7.5 cells expressing IFIT1 point mutants.

The online version of this article includes the following source data for figure 2:

**Source data 1.** Original images of membranes corresponding to *Figure 2*, panel D.

**Source data 2.** Original files for western blot analysis displayed in *Figure 2*.

human IFIT1 and span 11 orders within the superorders Euarchontoglires, Laurasiatheria, Afrotheria, and Xenarthra (*Figure 3A–B*; *Supplementary file 1*). All primate, bat, and ungulate orthologs in this screen were also part of the PAML, FUBAR, and MEME analyses, allowing us to correlate functional differences in antiviral activity with molecular evolutionary analysis. Each construct carried an N-terminal HA tag for estern blot detection. Huh7.5 cells transduced with these vectors were challenged with GFP-expressing VEEV or vesicular stomatitis virus (VSV). These viruses were selected because human IFIT1 restricts them by distinct mechanisms. VSV, a rhabdovirus with a negative-sense RNA genome, is inhibited by IFIT1 binding to its 5′-triphosphate RNA (5′-ppp-N), which sequesters the genome from the replication pool (*Pichlmair et al., 2011*). VEEV TC-83 is inhibited by IFIT1-mediated recognition of its Cap0 RNA, which leads to inhibition of translation (*Hyde et al., 2014*; *Reynaud et al., 2015*). IFIT1-expressing cells were infected with an MOI of 1 for both VSV-GFP and VEEV-GFP, and infection was quantified by flow cytometry (*Figure 3C*). Orthologs that strongly reduced infectivity by at least 70% relative to control cells transduced with an empty lentiviral vector were considered hits. Eleven IFIT1 orthologs (African savannah elephant, Chinese pangolin, sheep, white-tailed deer, black flying fox, dromedary camel, orca, Angola colobus, big brown bat, human, and large flying fox) inhibited VEEV (*Figure 3C*), and five (orca, cape elephant shrew, dolphin, European rabbit, and nine-banded armadillo) inhibited VSV (*Figure 3D*). We were surprised that hits for each virus spanned diverse clades with minimal overlap. Only orca IFIT1 restricted both viruses, and just 10 of 39 orthologs met the 70% inhibition threshold, indicating that multiple IFIT1 orthologs target neither VEEV or VSV under these experimental conditions. Western blots revealed wide variation in protein abundance (*Figure 3—figure supplement 1*), from high expression (e.g. orca, European rabbit) to very low levels (e.g. African green monkey, greater horseshoe bat). Black-capped squirrel monkey IFIT1 also migrated at a lower-than-expected mass, suggesting a truncation or cleavage of the protein product. However, several low-expressing orthologs were potent restrictors, whereas many highly expressed ones were inactive. These data suggest that IFIT1 orthologs may exhibit species-specific antiviral functions independent of differences in protein expression.

We next sought to validate the antiviral phenotypes of a subset of orthologs with more comparable expression. We selected nine orthologs based on protein expression, evolutionary diversity, and variable phenotypes in the screen, including IFIT1 from Chinese pangolin (hereafter referred to as pangolin), orca, sheep, black flying fox, big brown bat, chinchilla, human, chimpanzee, and cape elephant shrew (hereafter referred to as shrew) (*Figure 4A*). Huh7.5 cells were transduced with different doses of lentiviral vectors in an attempt to equalize expression. Western blots showed that eight orthologs were expressed at levels approximately fourfold above to twofold below human IFIT1. The notable exception was orca IFIT1, which was expressed approximately 15 times higher than human IFIT1 (*Figure 4B*). IFIT1-expressing cells were challenged with VEEV (*Figure 4C*) or VSV (*Figure 4D*) at an MOI of two. VSV was potently inhibited by orca and shrew IFIT1, while no other tested orthologs significantly suppressed infection. Thus, VSV may evade suppression by many mammalian IFIT1s. VEEV was potently inhibited by pangolin, sheep, black flying fox, big brown bat, and human IFIT1, while orca IFIT1 was modestly inhibitory. VEEV was not significantly inhibited by chimpanzee IFIT1 but was suppressed by human IFIT1. This was particularly noteworthy as human and chimpanzee IFIT1 protein sequences differ by only eight amino acids, and one of these (residue 366) was identified as rapidly evolving by both FUBAR and MEME (*Figure 1D*). Together, these validation experiments demonstrate that mammalian IFIT1 exhibits species-specific differences in viral suppression. Furthermore, while expression differences in IFIT1 orthologs are noted and may affect antiviral potency, our data suggest that species-specific differences cannot be fully explained by expression differences. This is exemplified by potent restriction of VEEV by pangolin and big brown bat IFIT1 despite low expression (*Figure 4B and C*). Furthermore, high-expressing sheep IFIT1 only restricted VEEV infection, but not VSV, and orca IFIT1 only modestly reduced VEEV infection despite much higher expression than all other IFIT1s (*Figure 4B–D*).

We next tested this subset of nine IFIT1 orthologs against human parainfluenza virus type 3 (PIV3). PIV3 is a paramyxovirus that, like the rhabdovirus VSV, has a negative-sense single-stranded RNA genome, allowing us to compare each ortholog's activity against two viruses that share a replication strategy. We also selected PIV3 because published reports differ on its susceptibility to human IFIT1, whereas the related PIV5 is consistently IFIT1-sensitive (*Young et al., 2016*; *Rabbani et al., 2016*; *Hankinson et al., 2025*; *Andrejeva et al., 2013*). PIV3 was significantly restricted by shrew, pangolin,

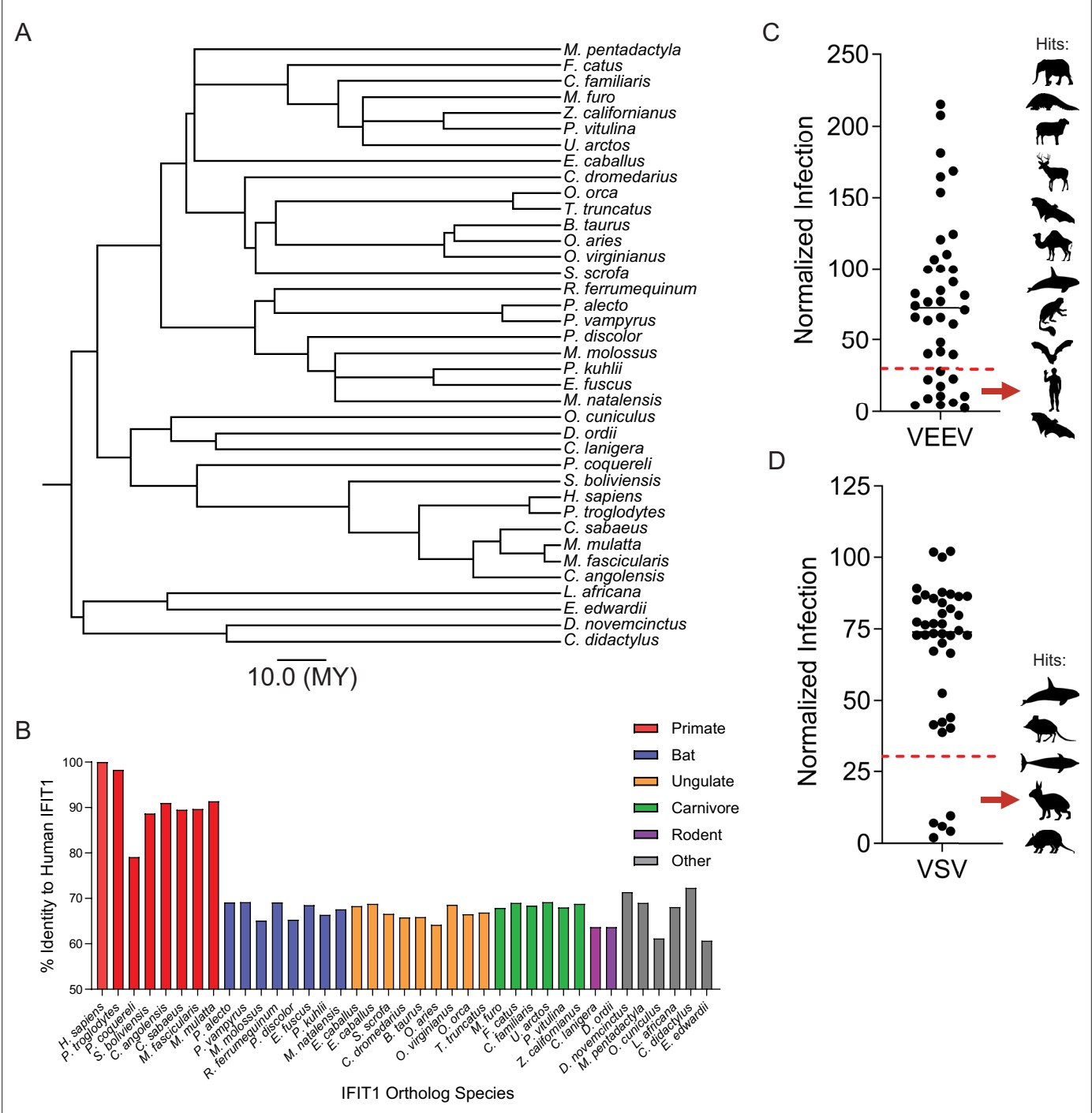

**Figure 3.** Interferon-induced protein with tetratricopeptide repeats 1 (IFIT1) ortholog screens reveal extensive heterogeneity in mammalian IFIT1 antiviral function. (**A**) TimeTree illustrating the evolutionary relationship of IFIT1 orthologs selected for ortholog screen. Scale bar represents divergence time of 10 million years. (**B**) Graph of protein sequence identity of IFIT1 orthologs used in screen relative to human IFIT1. (**C**) Dot plot representing the relative infection (compared to control cells) from an ectopic overexpression screen in which Huh7.5 cells expressing 39 different IFIT1 mammalian orthologs were challenged with 1.0 MOI VEEV-GFP for 4 hr. Infectivity was quantified by flow cytometry. Red line denotes 30% relative infection. n=2 independent experiments. (**D**) Same as C, for VSV-GFP (1 MOI, 4 hr infection). n=3 independent experiments. In C, D, silhouettes represent the species with a relative inhibition of at least 70% in order from most (top) to least (bottom) inhibitory.

The online version of this article includes the following source data and figure supplement(s) for figure 3:

**Figure supplement 1.** Interferon-induced protein with tetratricopeptide repeats 1 (IFIT1) ortholog expression varied during the ortholog screen.

*Figure 3 continued on next page*

Figure 3 continued

**Figure supplement 1—source data 1.** Original images of membranes corresponding to *Figure 3—figure supplement 1A*.

**Figure supplement 1—source data 2.** Original files for western blot analysis displayed in *Figure 3—figure supplement 1A*.

chinchilla, black flying fox, and big brown bat IFIT1 (*Figure 4E*). Suppression of PIV3 by human IFIT1 in our assay was very modest and not statistically significant. Chinchilla IFIT1 exhibited disparate activities, as it did not inhibit VEEV (*Figure 4C*) or VSV (*Figure 4D*), yet significantly protected cells from PIV3 infection (*Figure 4E*). We then tested these IFIT1 orthologs against another alphavirus related to VEEV, Sindbis virus (SINV) (*Figure 4E*). SINV was largely resistant to most mammalian IFIT1s, consistent with previous human IFIT1 studies (*Hyde et al., 2014*; *Reynaud et al., 2015*). However, SINV was potently suppressed by orca IFIT1, and significantly suppressed by pangolin, sheep, and chinchilla IFIT1 (*Figure 4E*), highlighting a pattern of suppression that was distinct across the nine orthologs relative to the alphavirus VEEV. Together, these data demonstrate that IFIT1 orthologs exhibit a range of antiviral phenotypes with a panel of genetically diverse viruses (*Figure 4G*), largely independent of differences in protein expression.

## Human, black flying fox, and chimpanzee IFIT1 protein exhibit species-specific RNA binding

Human and black flying fox IFIT1 inhibited VEEV, whereas chimpanzee IFIT1 did not (*Figure 4C*). The black flying fox (*Pteropus alecto*) is a model megabat species that is a reservoir for zoonotic viruses (*Halpin et al., 2000*; *Marsh et al., 2012*; *Barr et al., 2012*) and expresses restriction factors with unique antiviral activities (*Cruz-Rivera et al., 2024*; *Boys et al., 2020*; *Hayward et al., 2018*; *Hayward et al., 2022*; *Morrison et al., 2020*; *Ohkura et al., 2023*; *Zhang et al., 2017*). While human and black flying fox IFIT1 share only 69% identity, human and chimpanzee IFIT1 are over 98% identical (*Figure 3B*, *Supplementary file 1*). The unique phenotypes of these three species, combined with their well-annotated genomes and the ease of purifying IFIT1 in bacteria, prompted us to use them for biochemical studies to examine potential differences in their RNA-binding ability.

Recombinant IFIT1 proteins (*Figure 3—figure supplement 1*) were tested for Cap0 RNA binding by electrophoretic mobility shift assay (EMSA). A 41-nt transcript corresponding to the VEEV 5′ UTR was capped with faustovirus capping enzyme to generate RNAs that resemble physiologic Cap0 termini (*Chan et al., 2023*). After incubation with purified IFIT1 orthologs, protein–RNA complexes were resolved by native PAGE and visualized (*Figure 5A*). Quantification of band shift intensity and area under the curve analysis revealed that black flying fox IFIT1 exhibited the strongest RNA binding activity, followed by human IFIT1 (*Figure 5B and C*). Little binding was observed by chimpanzee IFIT1, as band shift intensity was barely above that of a human RNA-binding deficient control (human IFIT1-R187H) (*Pichlmair et al., 2011*; *Abbas et al., 2017*; *Figure 5B and C*). These biochemical data are consistent with viral phenotypes (*Figure 4C*), suggesting that species-specific antiviral activities of IFIT1s can be partly explained by RNA-binding potential.

## Mutagenesis of primate IFIT1 reveals genetic determinants of IFIT1 antiviral function

Human IFIT1 restricts VEEV and binds its Cap0 RNA, whereas chimpanzee IFIT1 shows no VEEV inhibition and little VEEV Cap0 RNA binding. The two proteins differ at only eight residues (*Figure 6A*). We, therefore, used site-directed mutagenesis to pinpoint which substitutions account for this functional gap. Because three of the eight variant residues—362, 364, and 366—clustered together (*Figure 6A*), we created reciprocal '362/4/6' triple-mutant IFIT1 constructs that swap these positions between chimpanzee and human proteins. One of these (human L366, chimpanzee M366) shows positive selection in two computational models (*Figure 1D*), implying that evolutionary pressure may have shaped the functional divergence between these primate IFIT1 orthologs.

Huh7.5 cells expressing wild-type (WT) or mutant IFIT1 were challenged with VEEV-GFP at an MOI of two, and infectivity was quantified by flow cytometry (*Figure 6B*). Western blotting demonstrated near uniform expression of all IFIT1 proteins (*Figure 6B*). WT chimpanzee IFIT1 conferred little protection (approximately 50% infection versus 60% in empty-vector controls), whereas WT human IFIT1 reduced infection to less than 10%. In the human background, only the L366M substitution caused a

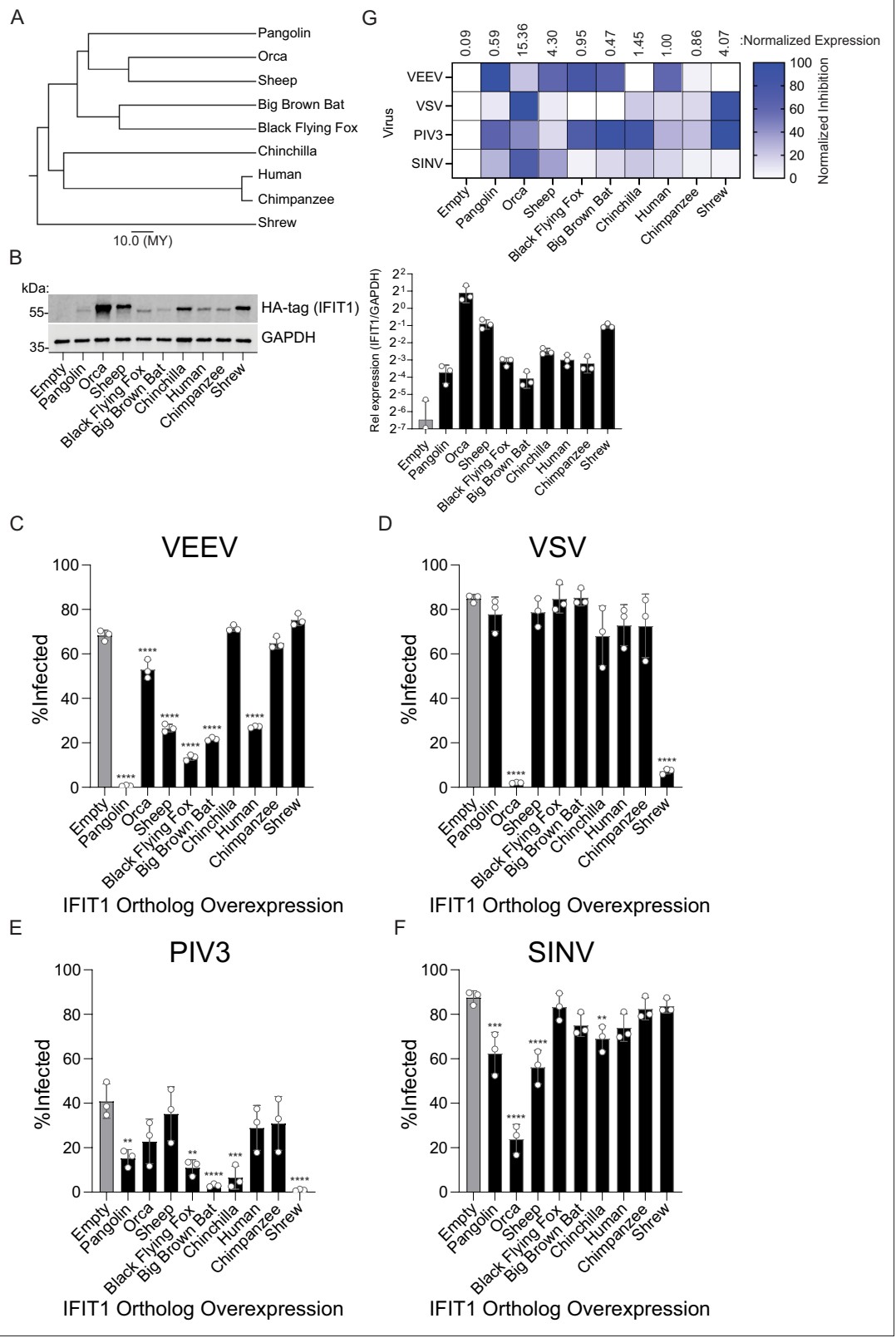

**Figure 4.** Validation of selected Interferon-induced protein with tetratricopeptide repeats 1 (IFIT1s) from ortholog screen. (**A**) TimeTree illustrating the evolutionary relationship of IFIT1 orthologs selected for screen validation and follow-up. Scale bar represents divergence time of 10 million years. (**B**) Western blot from cells expressing HA-tagged IFIT1 orthologs used in C-F. Image is representative of three independent replicates. Quantification

*Figure 4 continued on next page*

*Figure 4 continued*

of band intensity was performed using LiCOR Image Studio C. Infection of IFIT1 ortholog-expressing Huh7.5 cells with VEEV-GFP (MOI 2, 4 hr) (**D**) Infection of IFIT1 ortholog-expressing Huh7.5 cells with VSV-GFP (MOI 2, 4 hr). (**E**) Infection of IFIT1 ortholog-expressing Huh7.5 cells with PIV3-GFP (MOI 2, 10 hr). (**F**) Infection of IFIT1 ortholog-expressing Huh7.5 cells with SINV-GFP (MOI 2, 10 hr). (**G**) Heat map summarizing the infection data shown in C-F and complementary ortholog expression data shown in B. In C-F, data represent mean ± SD, n=3 independent experiments. Statistical significance was determined by one-way ANOVA with Dunnett's test. ns p>0.05, **p<0.01, ***p<0.001, and ****p<0.0001.

The online version of this article includes the following source data for figure 4:

**Source data 1.** Original images of membranes corresponding to *Figure 4B*.

**Source data 2.** Original files for western blot analysis displayed in *Figure 4B*.

---

significant loss of restriction (approximately fourfold higher infection than WT). Q364R had a modest, non-significant effect, and the 362/4/6 triple mutant nearly abolished activity. In the chimpanzee background, the reciprocal M366L substitution reduced infection by half, R364Q produced a modest, non-significant gain in antiviral activity, and the 362/4/6 triple mutant suppressed VEEV as effectively as WT human IFIT1. These results show that residues N362, Q364, and L366 are necessary for VEEV restriction by human IFIT1, and that their reciprocal substitution is sufficient to endow chimpanzee IFIT1 with potent antiviral activity.

Amino acids 362, 364, and 366 are generally conserved across 20 primate IFIT1 sequences (*Figure 6C*). Residue 366 is typically hydrophobic. Fifteen species, including humans, encode leucine, whereas the other five, including chimpanzees, encode either valine or methionine (*Figure 6C*). This suggests that leucine at position 366, which is rapidly evolving, might be important for antiviral function in primate IFIT1. Additional support for the importance of L366 comes from our findings that the only primate IFIT1, besides those of chimpanzees and humans, which was expressed at appreciable levels during the screen was that of the Angola colobus. This IFIT1 variant also potently inhibited VEEV infection and notably contains a leucine at residue 366, similar to human IFIT1 (*Figure 3C*, *Figure 3—figure supplement 1Supplementary file 1*). Glutamine is conserved at site 364 in all primate species aligned except chimpanzee, which encodes an arginine (Q364R) and introduces a positive charge. Point mutants of residue 364 in chimpanzee and human IFIT1 exhibited moderate alterations in anti-VEEV function, though not statistically significant, suggesting that this amino acid may have a minor role in antiviral restriction that synergizes with L366.

To test the combined effects of residues 364 and 366, double mutants (referred to as 364/6 double mut) for both chimpanzee and human IFIT1 were generated. Huh7.5 cells expressing WT, 364/6 double mut, or 362/4/6 triple mut IFIT1s of both chimpanzee and human IFIT1 were challenged with VEEV-GFP at an MOI of two, and infection was quantified by flow cytometry (*Figure 6D*). Western blotting demonstrated uniform expression of all human and chimpanzee IFIT1 mutants (*Figure 6D*). As in previous experiments, wild-type human IFIT1 strongly suppressed VEEV infection, whereas wild-type chimpanzee IFIT1 did not (*Figure 6D*). However, the 364/6 double mut and 362/4/6 triple mut of human IFIT1 both had a near complete loss of antiviral capacity (*Figure 6D*). These results suggest that residues 364 and 366, but not 362, in human IFIT1 are necessary to confer anti-VEEV function. In contrast, the 364/6 double mutant or the 362/4/6 triple mutant of chimpanzee IFIT1 potently suppressed VEEV infection similar WT human IFIT1. No difference in antiviral potency was observed between chimpanzee IFIT1 with the 364/6 double mut or 362/4/6 triple mut, suggesting again that residue 362 does not play a key role in the differential antiviral properties of human and chimpanzee IFIT1. Together, these data indicate that out of eight amino acids that differ between human and chimpanzee IFIT1, sites 364 and 366, in combination, comprise the underlying differences in anti-VEEV function.

We next examined the location of amino acids 364 and 366 in the solved crystal structure of human IFIT1 bound to RNA and in the chimpanzee IFIT1 structure predicted by AlphaFold. Residues 364 and 366 are located within the α-helix 18 within TPR7 of IFIT1 (*Figure 6A and E*). They are not predicted to contact RNA and are not located in direct proximity of the RNA-binding pocket (*Figure 6E and F*), suggesting they may regulate primate IFIT1 function in an allosteric manner.

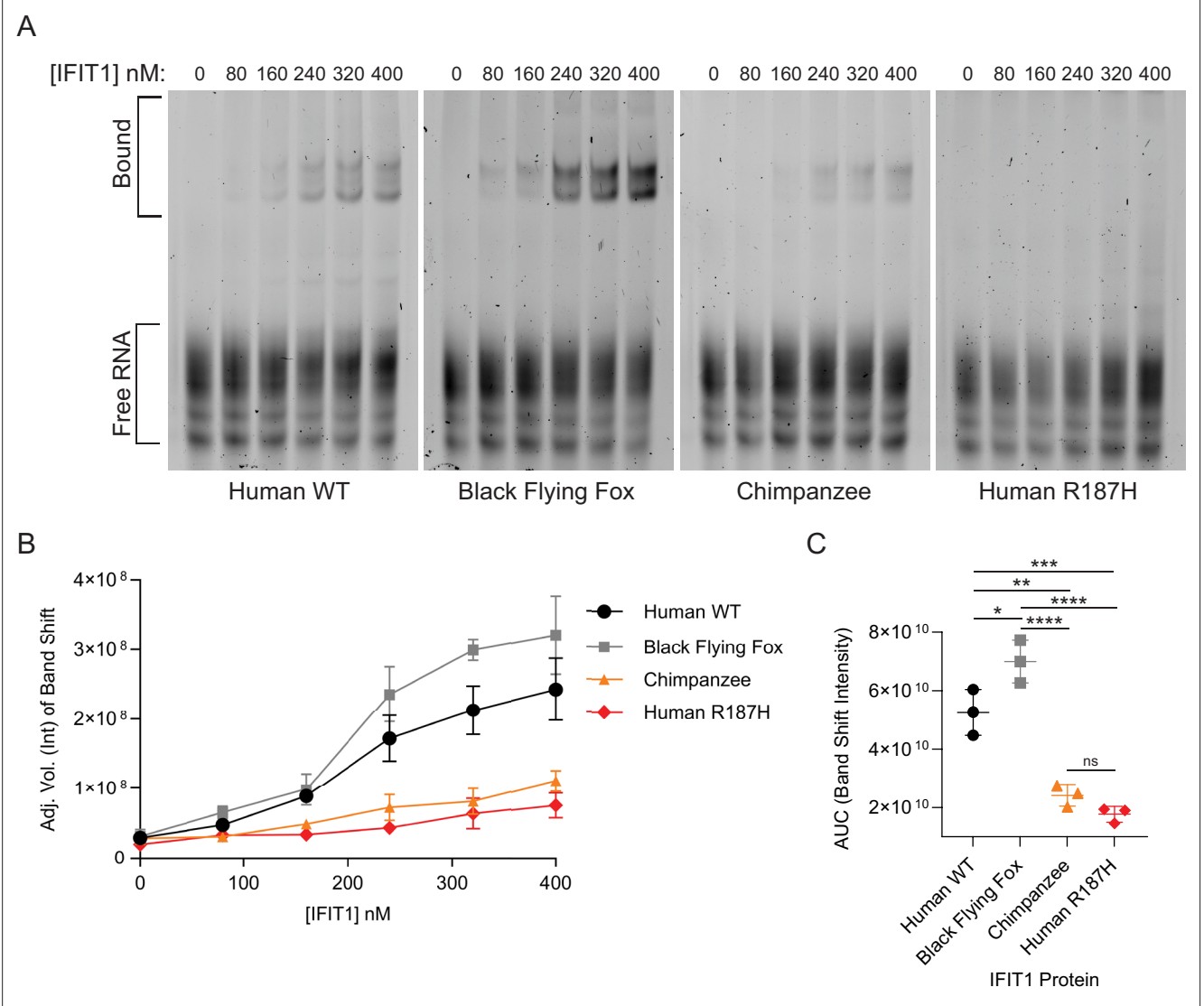

**Figure 5.** Interferon-induced protein with tetratricopeptide repeats 1 (IFIT1) proteins exhibit species-specific Cap0 RNA binding. (**A**) RNA electrophoretic mobility shift assay (EMSA) with Cap0 Venezuelan equine encephalitis virus (VEEV) RNA probes (41 nt of VEEV TC-83 strain 5' UTR) at 50 nM incubated with increasing concentrations of the indicated purified IFIT1 proteins. Images are one representative image from three independent replicates. (**B**) Plot of the band shift intensity from EMSAs in A. Data represent mean ± SD, n=3 independent experiments. Band intensity was quantified by ImageLab software (BioRad). (**C**) Area Under the Curve (AUC) analysis calculated from raw data in B. Statistical significance was tested by one-way ANOVA with correction for multiple comparisons. Data represent mean ± SD, n=3 independent experiments. ns, $p>0.05$, $*p<0.05$, $**p<0.01$, $***p<0.001$, and $****p<0.0001$.

The online version of this article includes the following source data and figure supplement(s) for figure 5:

**Source data 1.** Original images of SYBR Gold-stained native gels corresponding to ***Figure 5A***.

**Source data 2.** Original files for electrophoretic mobility shift assay (EMSA) displayed in ***Figure 5A***.

**Figure supplement 1.** Gel image of recombinant Interferon-induced protein with tetratricopeptide repeats 1 (IFIT1) protein.

**Figure supplement 1—source data 1.** Original image of Coomassie-stained SDS-PAGE gel corresponding to ***Figure 5—figure supplement 1***.

**Figure supplement 1—source data 2.** Original file for SDS-PAGE gel displayed in ***Figure 5—figure supplement 1***.

## Discussion

Here, combining evolutionary and functional data, we conclude that mammalian IFIT1s are rapidly evolving viral restriction factors that exhibit species-specific antiviral activity. Differential antiviral activity of human and chimpanzee IFIT1 led to the unexpected discovery that two residues, 364 and

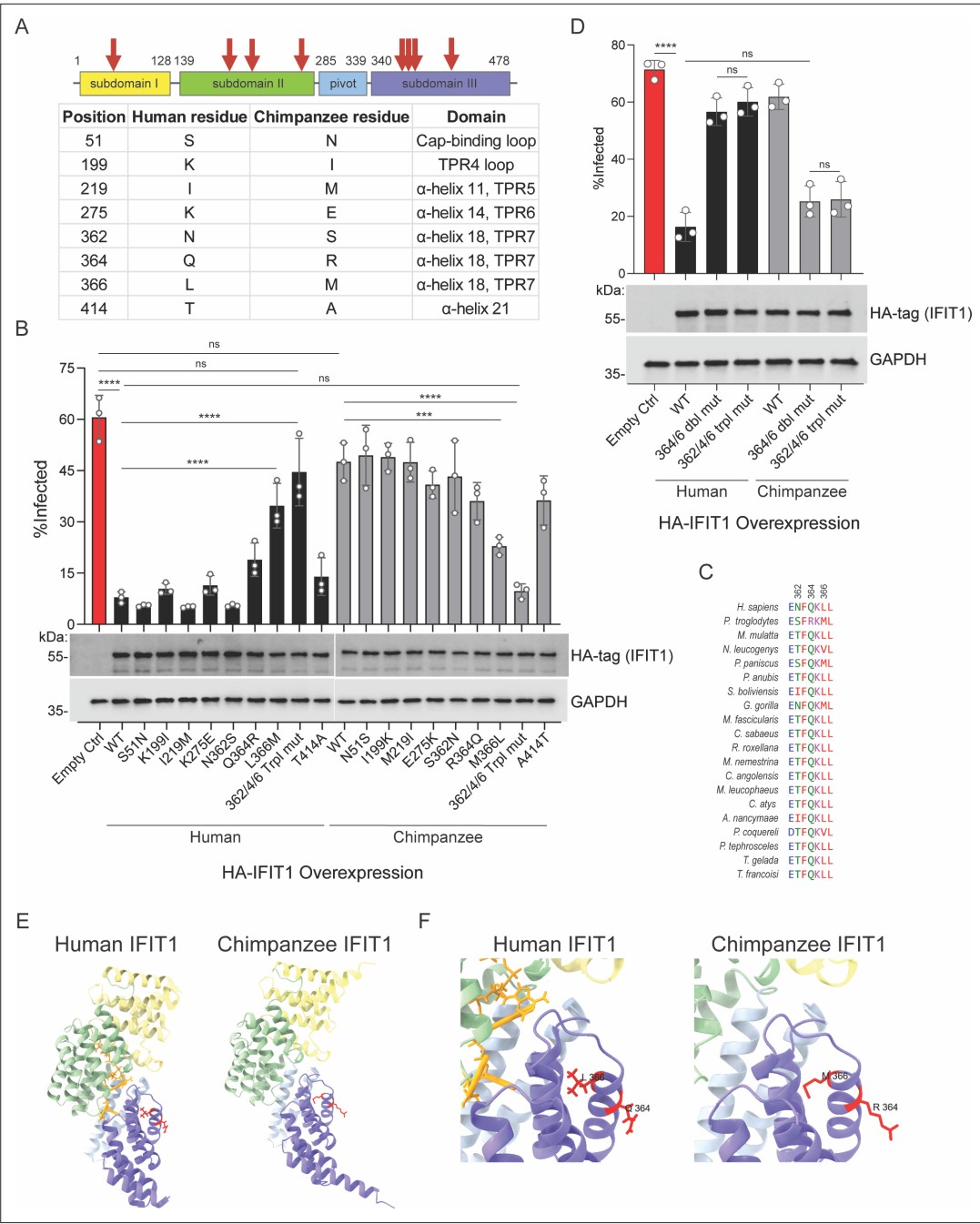

**Figure 6.** Mutagenesis uncovers genetic determinants of primate Interferon-induced protein with tetratricopeptide repeats 1 (IFIT1) antiviral function. (**A**) Diagram and chart describing amino acids that differ between human and chimpanzee IFIT1 as well as their domain location. (**B**) Effects of primate IFIT1 mutant expression on Venezuelan equine encephalitis virus (VEEV)-GFP infection. Huh7.5 cells were infected with an MOI of 2 for 4 hr and infectivity was quantified by flow cytometry. Data represent mean ± SD, n=3 independent experiments. Statistical significance was determined by one-way ANOVA with Dunnett's test. ns, p>0.05; ***p<0.001; and ****p<0.0001. Western blotting for HA-tag (IFIT1) and GAPDH (Loading control) was also performed to determine expression levels of IFIT1 mutants. (**C**) Clustal Omega protein sequence alignment of residues 361–367 of IFIT1 from 20 primate species. (**D**) Effects of primate IFIT1 double or triple mutants on VEEV-GFP infection. Huh7.5 cells were infected with an MOI of 2 for 4 hr and infectivity was quantified by flow cytometry. Data represent mean ± SD, n=3 independent experiments. Statistical significance was determined by one-way ANOVA with Dunnett's test. ns, p>0.05; and ****p<0.0001. Western blotting for HA-tag (IFIT1) and GAPDH (Loading control) was also performed to determine relative IFIT1 mutant protein abundance. (**E**) Structure of

*Figure 6 continued on next page*

*Figure 6 continued*

human IFIT1 bound to RNA (left) or AlphaFold predicted structure of chimpanzee IFIT1 (right) visualizing location of residues 364 and 366 (red) that confer antiviral activity. Graphics were generated using ChimeraX. (**F**) Zoom in of (**E**).

The online version of this article includes the following source data for figure 6:

**Source data 1.** Original images of membranes corresponding to *Figure 6B*, (top), and *Figure 6D*, (bottom).

**Source data 2.** Original files for western blot analysis displayed in *Figure 6*.

366, in α-helices 18 within the TPR7 domain of IFIT1 may partially determine primate IFIT1 antiviral capacity. The structures of human and chimp IFIT1 suggest that these amino acids are not directly involved in RNA binding. Thus, they may be involved in conformational changes of IFIT1 during RNA-binding or other allosteric effects that result in a natural loss of function, though additional studies are needed to test this.

Our data also support a model in which the breadth of IFIT1 antiviral phenotypes may be, at least partially, explained by differences in RNA-binding ability. VEEV infection was most potently suppressed by black flying fox IFIT1, significantly inhibited by human IFIT1, and unaffected by chimpanzee IFIT1 (*Figure 4C*). This trend was similarly observed in RNA-binding assays with a sequence identical to that of the Cap0 VEEV 5' UTR (*Figure 5*). These data are consistent with previous studies showing that the ability of IFIT1 to bind viral RNA is required for its antiviral function (*Diamond and Farzan, 2013*; *Fensterl and Sen, 2015*).

The differential antiviral activity between IFIT1B and IFIT1 across several mammals was previously demonstrated by *Daugherty et al., 2016*. This study used phylogenetic approaches to show that gene birth, gene loss, and gene conversion occurred throughout vertebrate *IFIT* genes, resulting in extensive genetic diversity (*Daugherty et al., 2016*). Furthermore, through viral infection assays and a yeast growth assay, the authors showed different RNA-binding properties and disparate antiviral functions between IFIT1 or IFIT1B proteins from primates, mouse, and cat (*Daugherty et al., 2016*). Our studies significantly expand the diversity of mammalian IFIT1s interrogated for antiviral function. Notably, IFIT1s across *Mammalia* are not universally antiviral against VEEV and VSV. In fact, antiviral specificity of IFIT1 orthologs is possibly quite narrow and highly dependent on which virus is being evaluated.

We further show that mammalian IFIT1s spanning diverse orders are rapidly evolving, and that in primates, this is occurring at functionally relevant sites that are not directly implicated in RNA binding. Computational studies identified amino acid 193 in TPR4 loop as rapidly evolving, and scanning mutagenesis showed that this residue is mutationally resilient. This was similarly observed for TRIM5α, in which most mutations within the v1 loop did not affect antiviral potency (*Tenthorey et al., 2020*). Alteration of charge within the v1 loop of TRIM5α did play a significant role in determining the antiviral outcome of TRIM5α mutants (*Tenthorey et al., 2020*). Similarly, the introduction of negatively charged residues at position 193 of human IFIT1 ablated antiviral activity (*Figure 2D*). Thus, alteration of this single amino acid within IFIT1 may not result in major gain or loss of antiviral function, which may maximize the potential for adaptation during evolutionary arms races with viruses. However, we cannot rule out that substitutions at position 193 may influence antiviral function when combined with changes at other sites in IFIT1. Position 193 is also under selection in bats, and multiple codons within or adjacent to the TPR4 loop register as positively selected across all clades examined (*Figure 1A–C*). Systematic mutagenesis of these residues—alone and in combination—will clarify whether specific side chains or overall loop charge drive species-specific antiviral outcomes.

We note several limitations of our study. [1] All IFIT1 ortholog screens were performed by expressing IFIT1 from distinct species in human cells, which may affect natural antiviral function. For example, some IFIT1 orthologs may bind viral RNA, but not interface with human proteins to effectively suppress translation, resulting in no antiviral activity when expressed in human cells. [2] Our experiments focus on expression of IFIT1 alone, without co-expression of other IFIT proteins. As human IFIT1 antiviral activity and RNA-binding is enhanced by interactions with other IFITs (*Mears and Sweeney, 2018*; *Geng et al., 2024*), it is possible that the activity of IFIT1 orthologs used in this study is also modulated by co-expression with other IFITs from the cognate species. In particular, human IFIT3 modulates IFIT1 Cap0 RNA-binding and antiviral activity (*Johnson et al., 2018*). Therefore, future experiments

testing the antiviral potential of co-expressed IFITs from diverse species would be informative. [3] Our biochemical assays focus on IFIT1 binding to Cap0 RNA only. This is due to comparative studies that have demonstrated IFIT1 has the strongest affinity for Cap0 RNA when compared to other RNAs, such as ppp-RNA (*Abbas et al., 2017*). This allowed us to use human IFIT1 as a positive control for Cap0 RNA-binding with a strong signal-to-noise for comparison to other IFIT1 orthologs. However, it is possible that differences in ppp-RNA binding may underlie differences in the antiviral potency of IFIT1 towards viruses such as VSV and PIV3. Furthermore, viral RNA secondary structure can affect IFIT binding (*Hyde et al., 2014*; *Hankinson et al., 2025*); the capacity of IFIT1 orthologs to bind RNA 5′ termini may, therefore, vary with tolerance to structure-based occlusion. [4] This study also does not address the possibility that differential viral evasion of IFIT1 proteins contributes to species-specific antiviral activity. In fact, a recent study has identified species-specific sensitivity of IFIT1 proteins to viral evasion factors encoded by poxviruses (*Xie et al., 2025*). [5] We have only provided genetic but not biochemical or structural evidence for the underlying differences between chimpanzee and human IFIT1 RNA-binding and antiviral activity. By mutagenesis assays, we demonstrate that two amino acid changes confer antiviral activity between chimpanzee and human IFIT1, suggesting a role for residues 364 and 366 in primate IFIT1. Additional insight into the role of these residues might be gained from molecular dynamics simulations using the crystal structure of human IFIT1 bound to Cap0 RNA as a template. It is also possible that these residues in human IFIT1 may be involved in interactions with other proteins, such as IFIT3, which modulates IFIT1 antiviral activity and Cap0 RNA-binding (*Johnson et al., 2018*). Indeed, subdomain III of IFIT1 (alpha-helices 17–23), which contains residues 364 and 366 (part of alpha helix 18), showed partial protection from deuterium uptake by hydrogen-deuterium exchange with mass spectrometry (HDX-MS) during IFIT1-IFIT3 binding experiments (*Johnson et al., 2018*). However, residues 364 and 366 themselves were not predicted to directly interact with IFIT3, and to date, structural studies have not demonstrated that these residues are directly involved in IFIT1-IFIT3 interactions. Comparisons with the 364/366 double mutant might determine whether these substitutions alter conformational dynamics in a manner that disfavors IFIT1-IFIT3 interaction and/or RNA binding. [6] Our experiments with chimp IFIT1 suggest that amino acid changes at sites 364 and 366 may have resulted in a natural loss of function for this ortholog. However, we only tested four viruses in infection studies and one viral RNA sequence in RNA-binding studies. Therefore, it is possible that chimpanzee IFIT1 inhibits other viruses, such as those that infect chimpanzees in nature. Future studies may also explore this possibility and uncover other amino acid changes that have resulted in IFIT1 loss of function in certain species. Were a loss of IFIT1 function to occur in chimpanzees or other species, this may have also coincided with a gain of dominant antiviral function in another IFIT or complex of IFIT proteins. However, this has yet to be experimentally tested.

Summarily, our findings underscore the notion that unbiased screening, investigations of restriction factors from non-model mammalian species, and evolutionary-guided approaches may uncover novel insight into host-pathogen interactions.

## Materials and methods
### Cells and cell culture
Huh7.5 and 293T cells (from C. Rice, The Rockefeller University) were maintained in 'complete' DMEM: DMEM (Gibco) supplemented with 10% FBS and 1x non-essential amino acids (NEAA; Gibco). All derived stable cells expressing selectable markers were grown in complete DMEM containing puromycin (Gibco; Huh7.5: 4 µg/mL). All cells were incubated at 37 °C in 5% $CO_2$. All cell lines have been STR profiled and confirmed negative for mycoplasma contamination by PCR testing (Venor GeM mycoplasma detection kit, Sigma).

### Viruses and viral infections
Venezuelan equine encephalitis virus (TC-83 strain) infectious clone was obtained from I.Frolov. VEEV-GFP stocks were generated as previously described (*Atasheva et al., 2010*). Briefly, infectious clone plasmid was linearized, and RNA was in vitro transcribed with the mMESSAGE mMACHINE SP6 Transcription Kit (Invitrogen) followed by RNA clean-up with the MEGAclear Transcription Clean-Up Kit (Invitrogen). BHK-21J cells ($8×10^6$ cells) were then electroporated with 5 µg of in vitro transcribed RNA and plated. Media was changed 18 hr after plating, and the remaining supernatant was collected

8 hr post-media change. After centrifugation to remove cell debris, virus-containing supernatants were aliquoted and stored at –80 °C. Sindbis virus (SINV-GFP, clone S300 from M. Heise) (*Simmons et al., 2010*) was generated as previously described. Vesicular stomatitis virus (VSV-GFP) (from J. Rose) and human parainfluenza virus type 3 (PIV3-GFP) (JS strain, from Peter Collins) were propagated by passaging in BHK-21J cells and storing clarified supernatants as virus stocks at –80 °C. All viral infections were performed in DMEM supplemented with 1% FBS and 1 x non-essential amino acids at a total volume of 0.2 mL in 24-well tissue culture plates. After a 1 hr incubation period, 0.3 mL of complete DMEM was added to all wells until harvest. For analysis of viral infection by flow cytometry, cells were dissociated from the plate with 150 µL Accumax (Innovative Cell Technologies) and then fixed in paraformaldehyde at a final concentration of 1%. Cells were allowed to fix for at least 30 min at 4 °C and subsequently centrifuged at 1500 × *g* for 5 min. Supernatant was removed from cell pellets and fixed cells were then resuspended in FACS buffer (PBS [Gibco] supplemented with 3% FBS). Flow cytometry was performed on the Stratedigm S1000EON benchtop flow cytometer, with analysis in FlowJo software.

## Cloning and plasmids

IFIT1 coding sequences were obtained from the NCBI database. IFIT1 sequences were cloned into the Gateway expression vector pSCRPSY (*Kane et al., 2016*). IFIT1 orthologs were synthesized as gBlocks (IDT) with Gateway-compatible DNA sequences and an N-terminal HA tag. Human and chimpanzee IFIT1 point mutants were generated using site-directed mutagenesis PCR. Human and chimpanzee IFIT1 362/4/6 triple mutants and 364/6 double mutants were generated by Gibson cloning of gBlocks (IDT). For protein purification, IFIT1 protein coding sequences that were codon-optimized for bacterial expression were cloned into a modified pET28a bacterial expression vector, containing an N-terminal 6X-His tag followed by the yeast Sumo (smt3).

## Western blotting

Cells were lysed with M-PER (Mammalian Protein Extraction Reagent, Thermo Scientific) containing 1 x complete protease inhibitor cocktail (Roche) at 4 °C for 5 min with intermittent rocking. Lysate was then stored at –80 °C. For running SDS-PAGE, lysate was thawed on ice, and mixed to a final concentration of 1 x SDS Loading Buffer (0.2 M Tris-Cl pH 6.8, 5% SDS, 25% Glycerol, 0.025% Bromophenol Blue, and 6.25% beta-mercaptoethanol). Samples were run on 12% TGX FastCast acrylamide gels (Bio-Rad) or 4–20% Mini-PROTEAN TGX precast protein gels (BioRad) and transferred to nitrocellulose membranes using a TransBlot Turbo system (Bio-Rad). Membranes were blocked with 5% milk in 1 x Tris Buffered Saline, with Tween-20 (TBS-T [20 mM Tris, 150 mM NaCl, 0.1% Tween-20]) for at least 45 min. Subsequent incubation with primary antibody (at dilutions between 1:500 to 1:5000) diluted in 1% milk in TBS-T was performed overnight, rocking at 4 °C. Primary antibodies probing for GAPDH (Proteintech Cat# 10494–1-AP, RRID:AB_2263076 or Proteintech Cat# 60004–1-Ig, RRID:AB_2107436), IFIT1 (Proteintech Cat# 23247-1-AP, RRID:AB_2811269), and HA-tagged IFIT1 (Biolegend Cat# 901502, RRID:AB_2565006) were used in this study. After incubation with primary antibodies, membranes were washed 3 times with TBS-T by rocking for 5 min. Secondary antibodies (Goat anti-Rabbit 800CW and Goat anti-Mouse 680RD [LI-COR] at 1:5000 dilutions) diluted in 1 x TBS-T were incubated with membranes for at least 45 min. Membranes were then washed three times with TBS-T by rocking for 5 min. Proteins were visualized using the LI-COR Odyssey FC imaging system.

## Lentivirus production and transduction

To generate lentivirus pseudoparticles, 350,000 293Ts were plated onto poly-lysine-coated 6-well plates. Lentiviral packaging plasmids expressing VSV-g and Gag-pol were then mixed with SCRPSY plasmids at a ratio of 0.1 µg:0.5 µg:2.5 µg, respectively. Plasmids were incubated for 20 min with a mix of XtremeGENE9 transfection reagent (Roche) and OptiMEM (Gibco). Transfection complexes were then added to 293T cells in 1 mL of DMEM supplemented with 3% FBS dropwise. After 6 hr of incubation, media was changed to 1.5 mL of DMEM supplemented with 3% FBS. Supernatants containing lentiviral pseudoparticles were harvested at 48 and 72 hr post-transfection. Lentiviral pseudoparticles were then processed by adding 4 µg/mL Polybrene (Sigma), 20 mM HEPES, centrifuging at 1500 rpm for 5 min to remove cell debris, and storing at –80 °C until transductions.

Lentivirus transductions were performed in 6-well plates after plating 350,000 cells (Huh7.5). Final volume of transductions was either 1 mL or 1.5 mL of lentivirus mixed with 'pseudoparticle' DMEM (DMEM supplemented with 3% FBS, 4 µg/mL Polybrene, and 20 mM HEPES). After a 6 hr incubation at 37 °C, 2 mL of complete DMEM was added to all transductions. Transduced cells were split into selection media 48 hr post-transduction and passaged for a minimum of three passages in selection media.

## Protein purification

IFIT1 protein coding sequences that were codon-optimized for bacterial expression were cloned into a modified pET28a bacterial expression vector, containing an N-terminal 6X-His tag followed by the yeast Sumo (smt3). Plasmids were transformed into Rosetta DE3 cells. Transformants were grown in Terrific Broth (TB) with 50 mg/L Kanamycin and 34 mg/L of Chloramphenicol and Antifoam B at 37 °C. At the $OD_{600}$ of 0.4, the temperature was lowered to 18 °C. Protein expression was induced at $OD_{600}$=3 by addition of IPTG to 0.4 mM final concentration. Cells were harvested after an overnight incubation by centrifugation ($OD_{600}$=~8.5) and lysed in the lysis buffer (50 mM Tris-HCl pH 8, 500 mM NaCl, 15 mM Imidazole, 1 mM PMSF, 4 mM β-ME) by sonication. Cell lysates were cleared by centrifugation at 35,000 × g for 30 min. The cleared lysate was applied onto a gravity column with pre-washed Ni-NTA resin. Resin was washed with 50 mM Tris-HCl pH 8, 1 M NaCl, 30 mM Imidazole, 4 mM β-ME. Proteins were eluted with 50 mM Tris-HCl pH 8, 300 mM NaCl, 300 mM Imidazole, 2 mM DTT. Proteins were treated with Ulp1 protease overnight to remove the Sumo tag and passed over a size-exclusion column (HiLoad 16/600 Superdex 200 pg), equilibrated with 50 mM Tris 8.0, 150 mM NaCl, 1 mM DTT. Peak fractions were pooled, concentrated by ultrafiltration, and stored at –80 °C until use. For SDS-PAGE gels of purified IFIT1 protein, 500 nM of human, black flying fox, chimpanzee, and human R187H mutant IFIT1 were boiled in 4 X Laemmli Sample Buffer (BioRad) with 1.25% BME and loaded into a 4–20% Mini-PROTEAN TGX Precast Protein Gel (BioRad). Gels were run and subsequently incubated in Coomassie stain (40% ethanol, 10% glacial acetic acid, 0.1% Coomassie R-250) for 1 hr and destained overnight (10% ethanol, 7.5% glacial acetic acid). Gels were imaged with the ChemiDoc Imaging System (BioRad).

## RNA electrophoretic mobility shift assay

Protein stock concentration of IFIT1 orthologs was determined by diluting protein 1:5 in 1 x TBE and measuring A280. Extinction coefficient was determined by ProtParam tool (Expasy), and subsequently used to calculate and normalize protein concentrations between stocks.

Before loading, native gels (10% 19:1 Acrylamide: Bis-acrylamide, 25 mM NaCl, 1 x TBE, 0.0005% APS, 0.001% TEMED) were pre-run with 1 x TBE in a cold room for at least 2.5 hr. While gels were pre-running, binding reactions were set up at room temperature after thawing and keeping both RNA and protein stocks on ice. All dilutions of protein and RNA were in 1 x reaction buffer (50 nM TRIS pH 8.0, 100 mM NaCl, 1 mM EDTA, 0.01 mg/mL Heparin sodium sulfate, 2.5% Glycerol). For binding reactions, 50 nM of Cap0 RNA was mixed 1:1 with 0, 80, 160, 240, 320, and 400 nM of purified IFIT1 protein in a final volume of 20 µL and incubated at room temperature for 20 min.

18 µL of RNA:Protein complexes were then run on pre-run native gels for 2.5 hr at 100 V in 1 x TBE in a cold room. Gels were then stained with 1 x SYBR Gold stain (ThermoFisher Scientific) diluted in 1 x TBE for 20 min rocking at room temperature and imaged on a ChemiDoc Imaging System (BioRad) with UV transillumination. Band intensity was subsequently analyzed using ImageLab software (BioRad).

## In vitro transcription and 5' capping of RNA probes

Uncapped RNA probes were generated using the T7 RiboMAX Express Large Scale RNA Production System (Promega). Manufacturer instructions were modified for generating short RNA probes. Briefly, 10 µM of DNA oligos corresponding to the T7φ2.5 promoter sequence (5- TAATACGACTCACTAT TA-3) and the complementary T7φ2.5 promoter sequence followed by the first 41 nt of the VEEV TC-83 strain 5' UTR sequence (5- GGGTAGGTAATTGGTCTGGGCTTCTCTCATGCGCCGCCTATAAT AGTGAGTCGTATTA-3) were boiled at 95 °C for 2–3 min and then slow cooled to room temperature to allow annealing. 1 µM of the annealed DNA oligos served as the template per 20 µL T7

transcription reaction. RNA was purified according to the manufacturer instructions using phenol:chloroform nucleic acid extraction (Promega).

To generate Cap0 RNA, purified uncapped RNA probes were modified using the Faustovirus Capping Enzyme (FCE) (NEB) using modified manufacturer instructions. Briefly, 100 μL reactions were composed of 20 μL of uncapped RNA purified after T7 transcription (described above), 0.2 mM SAM, 1 mM GTP, 7500 U of FCE, 1 x FCE reaction buffer, 80 U RNasin Plus Ribonuclease Inhibitor (Promega), and nuclease-free water up to 100 μL. Reactions were incubated at 42 °C for 4.5 hr and then another 0.5 hr at 50 °C. Cap0 RNA was then purified using the same phenol:chloroform nucleic acid extraction as performed after the T7 transcription reaction (Promega). Cap0 RNA was determined to be at least 95% Cap0 RNA (m7GpppA) by targeted LC/MS after Nuclease P1 (NEB) digestion at 37 °C for 30 min followed by inactivation of enzyme at 95 °C for 2 min. LC/MS was performed by the UT Southwestern Preclinical Pharmacology Core Facility.

## Computational evolutionary analysis

IFIT1 coding sequences for all analyses were obtained from the NCBI database. Alignment of multiple sequences was performed using MUSCLE and ClustalW implemented in MEGA X (*Kumar et al., 2018*). Maximum-likelihood trees were constructed in MEGA X, with standard settings. Trimmed alignments were then generated (*Castresana, 2000*), and when required, ALTER (*Glez-Peña et al., 2010*) was used to convert file formats. MEME (*Murrell et al., 2015*) and FUBAR (*Murrell et al., 2013*) were performed with HyPhy (*Kosakovsky Pond et al., 2020*) using the Datamonkey web application (*Weaver et al., 2018*). To perform PAML analysis, we used default settings with the F3x4 codon frequency table within CodeML. Likelihood ratio tests were performed to compare model 7 versus model 8 and model 8 versus model 8 a to determine the presence of positive selection. Bayes empirical Bayes testing was performed to determine specific residues that are adaptively evolving. To detect potential recombination in IFIT1, GARD analysis (*Kosakovsky Pond et al., 2006*) was performed on aligned coding sequences with HyPhy (*Kosakovsky Pond et al., 2020*) using the Datamonkey web application (*Weaver et al., 2018*). Recombination Detection Program version 4 (RDP4) (*Martin et al., 2015*) was also used to remove inferred recombinant sequences from alignment, using default settings. Maximum likelihood tree comparing aligned IFIT1, IFIT1B, and IFIT3 C-terminal sequences was aligned using MUSCLE in MEGA X, followed by maximum likelihood tree generation with 1000 bootstrap replicates in MEGA X (*Kumar et al., 2018*). Alignment to the C-terminal region of IFITs to distinguish IFIT1 and IFIT1B sequences was previously described (*Daugherty et al., 2016*). Species trees were generated using the TimeTree database (*Kumar et al., 2017*). FigTree software was then used to make trees shown in figures. Alignments and determination of amino acid identity to human IFIT1 was performed using Clustal Omega (*Sievers et al., 2011*). Graphics comparing human and chimpanzee IFIT1 structures were generated using ChimeraX (*Meng et al., 2023*).

## Statistical analysis

All statistical analyses were performed in GraphPad Prism 9. All graphs represent the mean of n=3 independent biological replicates with error bars representing standard deviation. ns, $p>0.05$, * $p<0.05$, **$p<0.01$, ***$p<0.001$, and ****$p<0.0001$.

## Materials availability statement

Newly generated materials are available from the corresponding author upon reasonable request and completion of a standard material transfer agreement.

# Acknowledgements

We thank Adam Osinski and Gina Park in the lab of Vincent Tagliabracci for purifying IFIT1 protein and for technical assistance with RNA EMSAs. This work was funded in part by the National Science Foundation Graduate Research Fellowship (2019274212 to MBM) and grants to JWS: NIH Grant AI158124, The Welch Foundation (I-2013–20220331), the UT Southwestern High Impact / High Risk Grant, and an Investigator in the Pathogenesis of Infectious Disease Award from the Burroughs Wellcome Fund.

## Additional information

### Competing interests
John W Schoggins: Senior editor, *eLife*. The other authors declare that no competing interests exist.

### Funding

| Funder | Grant reference number | Author |
| --- | --- | --- |
| National Science Foundation | Graduate Research Fellowship Program 2019274212 | Matthew B McDougal |
| National Institutes of Health | AI158124 | John W Schoggins |
| Welch Foundation | I-2013-20220331 | John W Schoggins |
| Burroughs Wellcome Fund | PATH Award | John W Schoggins |

The funders had no role in study design, data collection and interpretation, or the decision to submit the work for publication.

### Author contributions
Matthew B McDougal, Conceptualization, Formal analysis, Funding acquisition, Investigation, Visualization, Methodology, Writing - original draft, Writing – review and editing; Ian N Boys, Formal analysis, Validation, Writing – review and editing; Anthony M De Maria, Emi Nakahara, Investigation; John W Schoggins, Conceptualization, Resources, Formal analysis, Supervision, Funding acquisition, Investigation, Visualization, Methodology, Writing - original draft, Project administration, Writing – review and editing

### Author ORCIDs
Matthew B McDougal ⓘ https://orcid.org/0000-0002-3957-8868
Ian N Boys ⓘ https://orcid.org/0000-0002-0854-207X
Emi Nakahara ⓘ https://orcid.org/0000-0001-9554-6759
John W Schoggins ⓘ https://orcid.org/0000-0002-7944-6800

Reviewer #2 (Public review): https://doi.org/10.7554/eLife.101929.3.sa1
Reviewer #3 (Public review): https://doi.org/10.7554/eLife.101929.3.sa2
Author response https://doi.org/10.7554/eLife.101929.3.sa3

## Additional files

### Supplementary files
MDAR checklist

Supplementary file 1. IFIT1 ortholog screen sequence information and viral infection data.

### Data availability
All data generated or analysed during this study are included in the manuscript and supporting files.

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
