## [Editor Report · eLife Assessment]

This **important** report describes the changing antiviral activity of IFIT1 across mammals and in response to distinct viruses, likely as a result of past arms races. One of the main strengths of the manuscript is the breadth of mammalian IFIT1 orthologs and viruses that were tested, as well as the thoroughness of the positive selection analysis. Overall the evidence is **convincing**, and the discussion conveys well the limitations due to physical interactions with other IFITs that are not accounted for.

---

## [Referee Report · Reviewer #2 (Public review)]

McDougal et al. describe the surprising finding that IFIT1 proteins from different mammalian species inhibit replication of different viruses, indicating that evolution of IFIT1 across mammals has resulted in host species-specific antiviral specificity. Before this work, research into the antiviral activity and specificity of IFIT1 had mostly focused on the human ortholog, which was described to inhibit viruses including vesicular stomatitis virus (VSV) and Venezuelan equine encephalitis virus (VEEV) but not other viruses including Sindbis virus (SINV) and parainfluenza virus type 3 (PIV3). In the current work, the authors first perform evolutionary analyses on IFIT1 genes across a wide range of mammalian species and reveal that IFIT1 genes have evolved under positive selection in primates, bats, carnivores, and ungulates. Based on these data, they hypothesize that IFIT1 proteins from these diverse mammalian groups may show distinct antiviral specificities against a panel of viruses. By generating human cells that express IFIT1 proteins from different mammalian species, the authors show a wide range of antiviral activities of mammalian IFIT1s. Most strikingly, they find several IFIT1 proteins that have completely different antiviral specificities relative to human IFIT1, including IFIT1s that fail to inhibit VSV or VEEV, but strongly inhibit PIV3 or SINV. These results indicate that there is potential for IFIT1 to inhibit a much wider range of viruses than human IFIT1 inhibits. Electrophoretic mobility shift assays (EMSAs) suggest that some of these changes in antiviral specificity can be ascribed to changes in direct binding of viral RNAs. Interestingly, they also find that chimpanzee IFIT1, which is >98% identical to human IFIT1, fails to inhibit any tested virus. Replacing three residues from chimpanzee IFIT1 with those from human IFIT1, one of which has evolved under positive selection in primates, restores activity to chimpanzee IFIT1. Together, these data reveal a vast diversity of IFIT1 antiviral specificity encoded by mammals, consistent with an IFIT1-virus evolutionary "arms race".

Overall, this is a very interesting and well-written manuscript that combines evolutionary and functional approaches to provide new insight into IFIT1 antiviral activity and species-specific antiviral immunity. The conclusion that IFIT1 genes in several mammalian lineages are evolving under positive selection is supported by the data. The virology results, which convincingly show that IFIT1s from different species have distinct antiviral specificity, are the most surprising and exciting part of the paper. As such, this paper will be interesting for researchers studying mechanisms of innate antiviral immunity, as well as those interested in species-specific antiviral immunity. Moreover, it may prompt others to test a wide range of orthologs of antiviral factors beyond those from humans or mice, which could further the concept of host-specific innate antiviral specificity. Additional areas for improvement, which are mostly to clarify the presentation of data and conclusions, are described below.

Strengths:

(1) This paper is a very strong demonstration of the concept that orthologous innate immune proteins can evolve distinct antiviral specificities. Specifically, the authors show that IFIT1 proteins from different mammalian species are able to inhibit replication of distinct groups of viruses, which is most clearly illustrated in Figure 4G. This is an unexpected finding, as the mechanism by which IFIT1 inhibits viral replication was assumed to be similar across orthologs. While the molecular basis for these differences remains unresolved, this is a clear indication that IFIT1 evolution functionally impacts host-specific antiviral immunity and that IFIT1 has the potential to inhibit a much wider range of viruses than previously described.

(2) By revealing these differences in antiviral specificity across IFIT1 orthologs, the authors highlight the importance of sampling antiviral proteins from different mammalian species to understand what functions are conserved and what functions are lineage- or species-specific. These results might therefore prompt similar investigations with other antiviral proteins, which could reveal a previously undiscovered diversity of specificities for other antiviral immunity proteins.

(3) The authors also surprisingly reveal that chimpanzee IFIT1 shows no antiviral activity against any tested virus despite only differing from human IFIT1 by eight amino acids. By mapping this loss of function to three residues on one helix of the protein, the authors shed new light on a region of the protein with no previously known function.

(4) Combined with evolutionary analyses that indicate that IFIT1 genes are evolving under positive selection in several mammalian groups, these functional data indicate that IFIT1 is engaged in an evolutionary "arms race" with viruses, which results in distinct antiviral specificities of IFIT1 proteins from different species.

Weaknesses:

(1) Some of the results and discussion text could be more focused on the model of evolution-driven changes in IFIT1 specificity. In particular, the majority of the residue mapping is on the chimpanzee protein, where it would appear that this protein has lost all antiviral function, rather than changing its antiviral specificity like some other examples in this paper. As such, the connection between the functional mapping of individual residues with the positive selection analysis and changes in antiviral specificity is not present. While the model that changes in antiviral specificity have been positively selected for is intriguing, it is not supported by data in the paper.

(2) The strength of the differences in antiviral specificity could be highlighted to a greater degree. Specifically, the text describes a number of interesting examples of differences in inhibition of viruses from Figure 3C and 3D, and 4C-F. The revised version has added some clarity by at least providing raw data for 3C and 3D for the reader to make their own comparisons, but it is still difficult to quickly assess which are the most interesting comparisons to make (e.g. for future mapping of residues that might be important).

---

## [Referee Report · Reviewer #3 (Public review)]

Summary:

This manuscript by McDougal et al, demonstrates species-specific activities of diverse IFIT1 orthologs, and seeks to utilize evolutionary analysis to identify key amino acids under positive selection that contribute to antiviral activity of this host factor. While the authors identify amino acid residues important for antiviral activity of some orthologs, and propose a possible mechanism by which these residues may function, the significance or applicability of these findings to other orthologs is unclear. However, the subject matter is of interest to the field, and these findings contribute to the body of knowledge regarding IFIT1 evolution.

Strengths:

Assessment of multiple IFIT1 orthologs shows the wide variety of antiviral activity of IFIT1, and identification of residues outside of the known RNA binding pocket in the protein suggests additional novel mechanisms which may regulate IFIT1 activity.

Weaknesses:

Given that there appears to be very little overlap observed in orthologs that inhibited the viruses tested, it's possible that other amino acids may be key drivers of antiviral activity in these other orthologs. Thus, it's difficult to conclude whether the findings that residues 362/4/6 are important for IFIT1 activity can be broadly applied to other orthologs, or whether these are unique to human and chimpanzee IFIT1. While additional molecular studies of the impact of these mutations on IFIT1 function (e.g. impact on IFIT complex formation) would lend further insight, as it stands, these findings demonstrate a role for these residues in IFIT1 activity.

---

## [Author Response]

The following is the authors’ response to the original reviews.

**Reviewer #1 (Public review):**
Summary:McDougal et al. aimed to characterize the antiviral activity of mammalian IFIT1 orthologs. They first performed three different evolutionary selection analyses within each major mammalian clade and identified some overlapping positive selection sites in IFIT1. They found that one site that is positively selected in primates is in the RNA-binding exit tunnel of IFIT1 and is tolerant of mutations to amino acids with similar biochemical properties. They then tested 9 diverse mammalian IFIT1 proteins against VEEV, VSV, PIV3, and SINV and found that each ortholog has distinct antiviral activities. Lastly, they compared human and chimpanzee IFIT1 and found that the determinant of their differential anti-VEEV activity may be partly attributed to their ability to bind Cap0 RNA.Strengths:The study is one of the first to test the antiviral activity of IFIT1 from diverse mammalian clades against VEEV, VSV, PIV3, and SINV. Cloning and expressing these 39 IFIT1 orthologs in addition to single and combinatorial mutants is not a trivial task. The positive connection between anti-VEEV activity and Cap0 RNA binding is interesting, suggesting that differences in RNA binding may explain differences in antiviral activity.Weaknesses:The evolutionary selection analyses yielded interesting results, but were not used to inform follow-up studies except for a positively selected site identified in primates. Since positive selection is one of the two major angles the authors proposed to investigate mammalian IFIT1 orthologs with, they should integrate the positive selection results with the rest of the paper more seamlessly, such as discussing the positive selection results and their implications, rather than just pointing out that positively selected sites were identified. The paper should elaborate on how the positive selection analyses PAML, FUBAR, and MEME complement one another to explain why the tests gave them different results. Interestingly, MEME which usually provides more sites did not identify site 193 in primates that was identified by both PAML and FUBAR. The authors should also provide the rationale for choosing to focus on the 3 sites identified in primates only. One of those sites, 193, was also found to be positively selected in bats, although the authors did not discuss or integrate that finding into the study. In Figure 1A, they also showed a dN/dS < 1 from PAML, which is confusing and would suggest negative selection instead of positive selection. Importantly, since the authors focused on the rapidly evolving site 193 in primates, they should test the IFIT1 orthologs against viruses that are known to infect primates to directly investigate the impact of the evolutionary arms race at this site on IFIT1 function.

We thank the reviewer for their assessment and for acknowledging the breadth of our dataset regarding diverse IFIT1s, number of viruses tested, and the functional data that may correlate biochemical properties of IFIT1 orthologous proteins with antiviral function. We have expanded the introduction and results sections to better explain and distinguish between PAML, FUBAR, and MEME analyses. Furthermore, we have expanded the discussion to incorporate the observation that site 193 is rapidly evolving in bats, as well as the observation that nearby sites to the TPR4 loop were identified as rapidly evolving in all clades of mammals tested. We also do observe an overall gene dN/dS of <1, however this is simply the average across all codons of the entire gene and does not rule out positive selection at specific sites. This is observed for other restriction factors, as many domains are undergoing purifying selection to retain core functions (e.g enzymatic function, structural integrity) while other domains (e.g. interfaces with viral antagonists or viral proteins) show strong positive selection. Specific examples include the restriction factors BST-2/Tetherin (PMID: 19461879) and MxA (PMID: 23084925). Furthermore, we agree that testing more IFIT1-sensitive viruses that naturally infect primates with our IFIT1 193 mutagenesis library would shed light on the influence of host-virus arms races at this site. However, VEEV naturally does also infect humans as well as at least one other species of primate (PMID: 39983680).

Below we individually address the reviewers' claims of inaccurate data interpretation.

Some of the data interpretation is not accurate. For example:(1) Lines 232-234: "...western blot analysis revealed that the expression of IFIT1 orthologs was relatively uniform, except for the higher expression of orca IFIT1 and notably lower expression of pangolin IFIT1 (Figure 4B)." In fact, most of the orthologs are not expressed in a "relatively uniform" manner e.g. big brown bat vs. shrew are quite different.

We have now included quantification of the western blots to allow the reader to compare infection results with the infection data (Updated Figure 4B and 4G). We have also removed the phrase “relatively uniform” from the text and have instead included text describing the quantified expression differences.

(2) Line 245: "...mammalian IFIT1 species-specific differences in viral suppression are largely independent of expression differences." While it is true that there is no correlation between protein expression and antiviral activity in each species, the authors cannot definitively conclude that the species-specific differences are independent of expression differences. Since the orthologs are clearly not expressed in the same amounts, it is impossible to fully assess their true antiviral activity. At the very least, the authors should acknowledge that the protein expression can affect antiviral activity. They should also consider quantifying the IFIT1 protein bands and normalizing each to GAPDH for readers to better compare protein expression and antiviral activity. The same issue is in Line 267.

We have now included quantification and normalization of the western blots to allow the reader to compare infection results with the infection data (Updated Figure 4B and 4G). Furthermore, we acknowledge in the text that expression differences may affect antiviral potency in infection experiments.

(3) Line 263: "SINV... was modestly suppressed by pangolin, sheep, and chinchilla IFIT1 (Figure 4E)..." The term "modestly suppressed" does not seem fitting if there is 60-70% infection in cells expressing pangolin and chinchilla IFIT1.

We have modified the text to say “significantly suppressed” rather than “modestly suppressed.”

(4) The study can be significantly improved if the authors can find a thread to connect each piece of data together, so the readers can form a cohesive story about mammalian IFIT1.

We appreciate the reviewer’s suggestion and have tried to make the story including more cohesive through commentary on positive selection and by using the computational analysis to first inform potential evolutionary consequences of IFIT1 functionality first by an intraspecies (human) approach, and then later an interspecies approach with diverse mammals that have great sequence diversity. Furthermore, we point out that almost all IFIT1s tested in the ortholog screen were also included in our computational analysis allowing for the potential to connect functional observations with those seen in the evolutionary analyses.

**Reviewer #2 (Public review):**
McDougal et al. describe the surprising finding that IFIT1 proteins from different mammalian species inhibit the replication of different viruses, indicating that the evolution of IFIT1 across mammals has resulted in host speciesspecific antiviral specificity. Before this work, research into the antiviral activity and specificity of IFIT1 had mostly focused on the human ortholog, which was described to inhibit viruses including vesicular stomatitis virus (VSV) and Venezuelan equine encephalitis virus (VEEV) but not other viruses including Sindbis virus (SINV) and parainfluenza virus type 3 (PIV3). In the current work, the authors first perform evolutionary analyses on IFIT1 genes across a wide range of mammalian species and reveal that IFIT1 genes have evolved under positive selection in primates, bats, carnivores, and ungulates. Based on these data, they hypothesize that IFIT1 proteins from these diverse mammalian groups may show distinct antiviral specificities against a panel of viruses. By generating human cells that express IFIT1 proteins from different mammalian species, the authors show a wide range of antiviral activities of mammalian IFIT1s. Most strikingly, they find several IFIT1 proteins that have completely different antiviral specificities relative to human IFIT1, including IFIT1s that fail to inhibit VSV or VEEV, but strongly inhibit PIV3 or SINV. These results indicate that there is potential for IFIT1 to inhibit a much wider range of viruses than human IFIT1 inhibits. Electrophoretic mobility shift assays (EMSAs) suggest that some of these changes in antiviral specificity can be ascribed to changes in the direct binding of viral RNAs. Interestingly, they also find that chimpanzee IFIT1, which is >98% identical to human IFIT1, fails to inhibit any tested virus. Replacing three residues from chimpanzee IFIT1 with those from human IFIT1, one of which has evolved under positive selection in primates, restores activity to chimpanzee IFIT1. Together, these data reveal a vast diversity of IFIT1 antiviral specificity encoded by mammals, consistent with an IFIT1-virus evolutionary "arms race".Overall, this is a very interesting and well-written manuscript that combines evolutionary and functional approaches to provide new insight into IFIT1 antiviral activity and species-specific antiviral immunity. The conclusion that IFIT1 genes in several mammalian lineages are evolving under positive selection is supported by the data, although there are some important analyses that need to be done to remove any confounding effects from gene recombination that has previously been described between IFIT1 and its paralog IFIT1B. The virology results, which convincingly show that IFIT1s from different species have distinct antiviral specificity, are the most surprising and exciting part of the paper. As such, this paper will be interesting for researchers studying mechanisms of innate antiviral immunity, as well as those interested in species-specific antiviral immunity. Moreover, it may prompt others to test a wide range of orthologs of antiviral factors beyond those from humans or mice, which could further the concept of host-specific innate antiviral specificity. Additional areas for improvement, which are mostly to clarify the presentation of data and conclusions, are described below.Strengths:(1) This paper is a very strong demonstration of the concept that orthologous innate immune proteins can evolve distinct antiviral specificities. Specifically, the authors show that IFIT1 proteins from different mammalian species are able to inhibit the replication of distinct groups of viruses, which is most clearly illustrated in Figure 4G. This is an unexpected finding, as the mechanism by which IFIT1 inhibits viral replication was assumed to be similar across orthologs. While the molecular basis for these differences remains unresolved, this is a clear indication that IFIT1 evolution functionally impacts host-specific antiviral immunity and that IFIT1 has the potential to inhibit a much wider range of viruses than previously described.(2) By revealing these differences in antiviral specificity across IFIT1 orthologs, the authors highlight the importance of sampling antiviral proteins from different mammalian species to understand what functions are conserved and what functions are lineage- or species-specific. These results might therefore prompt similar investigations with other antiviral proteins, which could reveal a previously undiscovered diversity of specificities for other antiviral immunity proteins.(3) The authors also surprisingly reveal that chimpanzee IFIT1 shows no antiviral activity against any tested virus despite only differing from human IFIT1 by eight amino acids. By mapping this loss of function to three residues on one helix of the protein, the authors shed new light on a region of the protein with no previously known function.(4) Combined with evolutionary analyses that indicate that IFIT1 genes are evolving under positive selection in several mammalian groups, these functional data indicate that IFIT1 is engaged in an evolutionary "arms race" with viruses, which results in distinct antiviral specificities of IFIT1 proteins from different species.Weaknesses:(1) The evolutionary analyses the authors perform appear to indicate that IFIT1 genes in several mammalian groups have evolved under positive selection. However, IFIT1 has previously been shown to have undergone recurrent instances of recombination with the paralogous IFIT1B, which can confound positive selection analyses such as the ones the authors perform. The authors should analyze their alignments for evidence of recombination using a tool such as GARD (in the same HyPhy package along with MEME and FUBAR). Detection of recombination in these alignments would invalidate their positive selection inferences, in which case the authors need to either analyze individual non-recombining domains or limit the number of species to those that are not undergoing recombination. While it is likely that these analyses will still reveal a signature of positive selection, this step is necessary to ensure that the signatures of selection and sites of positive selection are accurate.(2) The choice of IFIT1 homologs chosen for study needs to be described in more detail. Many mammalian species encode IFIT1 and IFIT1B proteins, which have been shown to have different antiviral specificity, and the evolutionary relationship between IFIT1 and IFIT1B paralogs is complicated by recombination. As such, the assertion that the proteins studied in this manuscript are IFIT1 orthologs requires additional support than the percent identity plot shown in Figure 3B.(3) Some of the results and discussion text could be more focused on the model of evolution-driven changes in IFIT1 specificity. In particular, the chimpanzee data are interesting, but it would appear that this protein has lost all antiviral function, rather than changing its antiviral specificity like some other examples in this paper. As such, the connection between the functional mapping of individual residues with the positive selection analysis is somewhat confusing. It would be more clear to discuss this as a natural loss of function of this IFIT1, which has occurred elsewhere repeatedly across the mammalian tree.(4) In other places in the manuscript, the strength of the differences in antiviral specificity could be highlighted to a greater degree. Specifically, the text describes a number of interesting examples of differences in inhibition of VSV versus VEEV from Figure 3C and 3D, but it is difficult for a reader to assess this as most of the dots are unlabeled and the primary data are not uploaded. A few potential suggestions would be to have a table of each ortholog with % infection by VSV and % infection by VEEV. Another possibility would be to plot these data as an XY scatter plot. This would highlight any species that deviate from the expected linear relationship between the inhibition of these two viruses, which would provide a larger panel of interesting IFIT1 antiviral specificities than the smaller number of species shown in Figure 4.

We thank the reviewer for their fair assessment of our manuscript. As the reviewer requested, we performed GARD analysis on our alignments used for PAML, FUBAR, and MEME (New Supp Fig 1). By GARD, we found 1 or 2 predicted breakpoints in each clade. However, much of the sequence was after or between the predicted breakpoints. Therefore, we were able to reanalyze for sites undergoing positive selection in the large region of the sequence that do not span the breakpoints. We were able to validate almost all sites originally identified as undergoing positive selection still exhibit signatures of positive selection taking these breakpoints into account: primates (11/12), bats (14/16), ungulates (30/37), and carnivores (2/4). To further validate our positive selection analysis, we used Recombination Detection Program 4 (RDP4) to remove inferred recombinant sequences from the primate IFIT1 alignment and performed PAML, FUBAR, and MEME. Once again, the sites in our original anlaysis were largely validated by this method. Importantly, sites 170, 193, and 366 in primates, which are discussed in our manuscript, were found to be undergoing positive selection in 2 of the 3 analyses using alignments after the indicated breakpoint in GARD and after removal of recombinant sequences by RDP4. We have updated the text to acknowledge IFIT1/IFIT1B recombination more clearly and include the GARD analysis as well as PAML, FUBAR, and MEME reanalysis taking into account predicted breakpoints by GARD and RDP4. Furthermore, to increase evidence that the sequences used in this study for both computational and functional analysis are IFIT1 orthologs rather than IFIT1B, we have included a maximum likelihood tree after aligning coding sequences on the C-terminal end (corresponding to bases 907-1437 of IFIT1). In Daughtery et al. 2016 (PMID: 27240734) this strategy was used to distinguish between IFIT1 and IFITB. All sequences used in our study grouped with IFIT1 sequences (including many confirmed IFIT1 sequences used in Daughterty et al.) rather than IFIT1B sequences or IFIT3. This new data, including the GARD, RDP4, and maximum likelihood tree is included as a new Supplementary Figure 1.

We also agree with the reviewer that it is possible that chimpanzee IFIT1 has lost antiviral function due to the residues 364 and 366 that differ from human IFIT1. We have updated the discussion sections to include the possibility that chimpanzee IFIT1 is an example of a natural loss of function that has occurred in other species over evolution as well as the potential consequences of this occurrence. Regarding highlighting the strength of differences in antiviral activity between IFIT1 orthologs, we have included several updates to strengthen the ability of the reader to assess these differences. First, we have included a supplementary table that includes the infection data for each ortholog from the VEEV and VSV screen to allow for readers to evaluate ranked antiviral activity of the species that suppress these viruses. In addition, the silhouettes next to the dot plots indicate the top ranked hits in order of viral inhibition (with the top being the most inhibitory) giving the reader a visual representation in the figure of top antiviral orthologs during our screen. We have also updated the figure legend to inform the reader of this information.

**Reviewer #3 (Public Review):**
Summary:This manuscript by McDougal et al, demonstrates species-specific activities of diverse IFIT1 orthologs and seeks to utilize evolutionary analysis to identify key amino acids under positive selection that contribute to the antiviral activity of this host factor. While the authors identify amino acid residues as important for the antiviral activity of some orthologs and propose a possible mechanism by which these residues may function, the significance or applicability of these findings to other orthologs is unclear. However, the subject matter is of interest to the field, and these findings could be significantly strengthened with additional data.Strengths:Assessment of multiple IFIT1 orthologs shows the wide variety of antiviral activity of IFIT1, and identification of residues outside of the known RNA binding pocket in the protein suggests additional novel mechanisms that may regulate IFIT1 activity.Weaknesses:Consideration of alternative hypotheses that might explain the variable and seemingly inconsistent antiviral activity of IFIT1 orthologs was not really considered. For example, studies show that IFIT1 activity may be regulated by interaction with other IFIT proteins but was not assessed in this study.Given that there appears to be very little overlap observed in orthologs that inhibited the viruses tested, it's possible that other amino acids may be key drivers of antiviral activity in these other orthologs. Thus, it's difficult to conclude whether the findings that residues 362/4/6 are important for IFIT1 activity can be broadly applied to other orthologs, or whether these are unique to human and chimpanzee IFIT1. Similarly, while the hypothesis that these residues impact IFIT1 activity in an allosteric manner is an attractive one, there is no data to support this.

We thank the reviewer for their fair assessment of our manuscript. To address the weaknesses that the reviewer has pointed out we have expanded the discussion to more directly address alternate hypotheses, such as the possibility of IFIT1 activity being regulated by interaction with other IFIT proteins. Furthermore, we expanded the discussion to include an alternate hypothesis for the role of residues 364 and 366 in primate IFIT1 besides allosteric regulation. In addition, we did not intend to claim or imply that residues 364/6 are the key drivers of antiviral activity for all IFITs tested. However, we speculate that within primates these residues may play a key role as these residues differ between chimpanzee IFIT1 (which lacks significant antiviral activity towards the viruses tested in this study) and human IFIT1 (which possesses significant antiviral activity). In addition, these residues seem to be generally conserved in primate species, apart from chimpanzee IFIT1. We have included changes to the text to more clearly indicate that we highlight the importance of these residues specifically for primate IFIT1, but not necessarily for all IFIT1 proteins in all clades.

**Reviewer #1 (Recommendations for the authors):**
(1) The readers would benefit from a more detailed background on the concept and estimation of positive selection for the readers, including the M7/8 models in PAML.

We have included more information in the text to provide a better background for the concepts of positive selection and how PAML tests for this using M7 and M8 models.

(2) Presentation of dataa) Figure 3C and 3D: is there a better way to present the infection data so the readers can tell the ranked antiviral activity of the species that suppress VEEV?

We have included a supplementary table that includes the infection data for each ortholog from the VEEV and VSV screen to allow for readers to evaluate ranked antiviral activity of the species that suppress these viruses. In addition, the silhouettes next to the dot plots indicate the top ranked hits in order of viral inhibition (with the top being the most inhibitory). We have updated the figure legend to inform the reader of this information as well.

b) Figure 4C and 4D: consider putting the western blot in Supplementary Figure 1 underneath the infection data or with the heatmap so readers can compare it with the antiviral activity.

We have also included quantification of the western blots performed to evaluate IFIT1 expression during the experiments shown in Figure 4C and 4D in an updated Figure 4B. We have also included normalized expression values with the heatmap shown in an updated Figure 4G so the reader can evaluate potential impact of protein expression on antiviral activity for all infection experiments shown in figure 4.

(3) Line 269-270: as a rationale for narrowing the species to human, black flying fox, and chimp IFIT1, human and black flying fox were chosen because they strongly inhibit VEEV, but pangolin wasn't included even though it had the strongest anti-VEEV activity?

The rationale for narrowing the species to human, black flying fox, and chimpanzee IFIT1 was related to the availability of biological tools, high quality genome/transcriptome sequencing databases, and other factors. Specifically human and chimp IFIT1 are closely related but have variable antiviral activities, making their comparison highly relevant. Bats are well established as reservoirs for diverse viruses, whereas the reservoir status of many other mammals is less well defined. Furthermore, purifying large amounts of high quality IFIT1 protein after bacterial expression was another limitation to functional studies. We have added this information into the manuscript text.

(4) Figure 5A: to strengthen the claim that "species-specific antiviral activities of IFIT1s can be partly explained by RNA binding potential", it would be good to include one more positive and one more negative control. In other words, test the cap0 RNA binding activity of an IFIT1 ortholog that strongly inhibits VEEV and an ortholog that does not. It would also be good to discuss why chimp IFIT1 still shows dose-dependent RNA binding yet it is one of the weakest at inhibiting VEEV.

We appreciate the reviewer's suggestion to include more controls and expand the dataset. While we understand the potential value of expanding the dataset, we believe that human IFIT1 serves as a robust positive control and human IFIT1 R187 (RNA-binding deficient) serves as an established negative control. Future experiments with other purified IFITs from other species will indeed strengthen evidence linking IFIT1 species-specific activity and RNA-binding.

Regarding chimpanzee IFIT1, we acknowledge there appears to be some dose-dependent Cap0 RNA-binding. However, the binding affinity is much weaker than that of human or black flying fox IFIT1. We speculate that during viral infection reduced binding affinity could impair the ability of chimpanzee IFIT1 to efficiently sequester viral RNA and inhibit viral translation. This reduction in binding affinity may, therefore, allow the cell to be overwhelmed by the exponential increase in viral RNA during replication resulting in an ineffective antiviral IFIT1. In the literature, a similar phenomenon is observed by Hyde et. al (PMID: 24482115). In this study, the authors test mouse Ifit1 Cap0 RNA binding by EMSA of the 5’ UTR sequence of VEEV RNA containing an A or G at nucleotide position 3. EMSA shows binding of both the A3 and G3 Cap0 VEEV RNA sequences, however stronger Ifit1 binding is observed for A3 Cap0 RNA sequence. The consequences of the reduced Ifit1 binding of the G3 Cap0 VEEV RNA are observed in vitro by a substantial increase in viral titers produced from cells as well as an increase in protein produced in a luciferase-based translation assay. The authors also show in vivo relevance of this reduction of Ifit1 binding as WT B6 mice infected with VEEV containing the A3 UTR exhibited 100% survival, while WT B6 mice infected with VEEV containing the G3 UTR survived at a rate of only ~25%. Therefore, the literature supports that a decrease in Cap0 RNA binding by an IFIT protein (while still exhibiting Cap0 RNA binding) observed by EMSA can result in considerable alterations of viral infection both in vitro and in vivo.

Minor:(1) Line 82: "including 5' triphosphate (5'-ppp-RNA), or viral RNAs..." having a comma here will make the sentence clearer.

We have improved the clarity of this sentence. It now reads, “IFIT1 binds uncapped 5′triphosphate RNA (5′-ppp-RNA) and capped but unmethylated RNA (Cap0, an m^7^G cap lacking 2′-O methylation).”

(2) Line 100: "...similar mechanisms have been at least partially evolutionarily conserved in IFIT proteins to restrict viral infection by IFIT proteins".

We have updated the text to improve clarity by revising the sentence to “VEEV TC-83 is sensitive to human IFIT1 and mouse Ifit1B, indicating at least partial conservation of antiviral function by IFIT proteins."

(3) Line 109: "signatures of rapid evolution or positive selection" would put positive selection second because that is the more technical term that can benefit from the more layperson term (rapid evolution).

We have updated this sentence incorporating this suggestion. “Positive selection, or rapid evolution, is denoted by a high ratio of nonsynonymous to synonymous substitutions (dN/dS >1).”

(4) Lines 116-117: "However, this was only assessed in a few species" would benefit from a citation.

We have inserted the citation.

(5) Line 127 heading: "IFIT1 is rapidly evolving in mammals" would be more accurate to say "in major clades of mammals".

We have updated the text to include this suggestion.

(6) Line 165: "IFIT1 L193 mutants".

We have updated the text to rephrase this for clarity.

(7) Line 170: two strains of VEEV were mentioned in the Intro, so it would be good to specify which strain of VEEV was used?

We have updated the text to clarify the VEEV strain. In this study, all experiments were performed using the VEEV TC-83 strain.

(8) Line 174: "Indeed, all mutants at position 193, whether hydrophobic or positively charged, inhibited VEEV similarly to the WT..." It should read "all hydrophobic and positively charged mutants inhibited VEEV similarly to the WT...".

We corrected as suggested.

(9) Line 204: what are "control cells"? Cells that are mock-infected, or cells without IFIT1?

We have updated the text to improve clarity. What we refer to as control cells, were cells expressing an empty vector control rather than an IFIT1.

(10) Need to clarify n=2 and n=3 replicates throughout the manuscript. Does that refer to three independent experiments? Or an experiment with triplicate wells/samples?

We have updated the text to say “independent experiments” instead of “biological replicates” to prevent any confusion. All n=2 or n=3 replicates denote independent experiments.

(11) Line 254: "dominant antiviral effector against the related human parainfluenza virus type 5..."

We have updated the text to improve clarity.

(12) Line 271: "The black flying fox (Pteropus alecto), is a model megabat species..." scientific name was italicized here but not elsewhere. Remove comma.

We have updated the text accordingly.

(13) Line 293: "...chimpanzee IFIT1 lacked these properties" but chimp IFIT1 can bind cap0 RNA, just at a lower level.

We have updated the text to acknowledge that chimpanzee IFIT1 can bind cap0 RNA, albeit at a lower level than human IFIT1.

(14) Figure 6B: please fix the x-axis labels. They're very cramped.

We have updated the x-axis labels for figure 6B and figure 6D to improve clarity.

(15) Line 609: "...trimmed and aligned"?

Our phrasing is to indicate that coding sequences were aligned, and gaps were removed to reduce the chance of false positive signal by underrepresented codons such as gaps or short insertions. We have removed “trimmed” from the text and changed the text to say “aligned sequences” to increase clarity.

**Reviewer #2 (Recommendations for the authors):**
(1) Numbers less than 10 should be spelled out throughout the manuscript (e.g. line 138).

We have updated the text to reflect the request.

(2) Line 165: "expression of IFIT1 193 mutants" should be rephrased.

We have updated the text to rephrase this sentence for clarity.

(3) A supplemental table or file should be included that contains the accession number and species names of sequences used for evolutionary analyses and for functional testing. In addition, the alignments that were used for positive selection can be included.

We have included a supplemental file containing accession numbers, species names for evolutionary analysis and functional studies. In addition, this table includes the infection data for each IFIT1 homolog for the screen performed in figure 3.

(4) The discussion of potential functions of the C-terminus of IFIT1 should include possible interactions with other proteins. In particular, the C-terminus of IFIT1 has been shown to interact with IFIT3 in a way that modulates its activity (PMID: 29525521). Although residues 362-366 were not shown in that paper to interact with a fragment of IFIT3, it is possible that these residues may be important for interaction with full-length IFIT3 or some other IFIT1 binding partner.

We thank the reviewer for their suggestion. We have expanded the discussion to explore the possibility that residues 364 and 366 of IFIT1 may be involved in IFIT1-IFIT3 interactions and consequently Cap0 RNA-binding and antiviral activity.

(5) The quantification of the EMSAs should be described in more detail. In particular, from looking at the images shown in Figure 5A, it would appear that human and chimpanzee IFIT1 show similar degrees of probe shift, while the human R187H panel shows no shifting at all. However, the quantification shows chimpanzee IFIT1 as being statistically indistinguishable from human R187H. Additional information on how bands were quantified and whether they were normalized to unshifted RNA would be helpful in attempting to resolve this visual discordance.

EMSAs were quantified by determining Adj. Vol. Intensity in ImageLab (BioRad), which subtracts background signal, after imaging at the same exposure and SYBR Gold staining time. To determine Adj. Vol. Intensity, we drew a box (same size for each gel and lane for each replicate) for each lane above the free probe. These values were not normalized to unshifted RNA, however equal RNA was loaded. While the ANOVA shows no significant difference, between human R187H and chimpanzee IFIT1 band shift intensity, this is potentially due to the between group variance in the ANOVA. The increase in the AUC value for chimpanzee IFIT1 is 36.4% higher than R187H.

The AUC of Adj. Vol. Intensity of human IFIT1 band shift is roughly 2-fold more than that of chimpanzee IFIT1. We believe this matches with the visual representation as well, as human IFIT1 has a darker “upper” band in the shift, as well as a clear dark “lower” band that is not well defined in the chimpanzee shift. Furthermore, the upper band of the chimpanzee IFIT1 shift appears to be as intense in the 400nM as the upper band in the 240nM human IFIT1 lane, without taking into account the lower band seen for human IFIT1 as well. We included this quantification as kD was unable to be calculated due to no clear probe disappearance and we do not intend for this quantification to act as a substitute for binding affinity calculations, rather to aid the reader in data interpretation.

**Reviewer #3 (Recommendations for the authors):**
(1) IFIT1 has been demonstrated to function in conjunction with other IFIT proteins, do you think the absence of antiviral activity is due to isolated expression of IFIT1 without these cofactors, and therefore might explain why there was little overlap observed in orthologs that inhibited the viruses tested (Figure 3, lines 209-210).

We do not believe that isolated expression of IFIT1 without cofactors (such as orthologous IFIT proteins) would fully explain the disparities in antiviral activity as many IFIT1s that expressed inhibited either VSV or VEEV in our screen. However, we acknowledge that the expression of IFIT1 alone does create a limitation in our study as IFIT1 antiviral activity and RNA-binding can be modulated by interactions with other IFIT proteins. Therefore, we do believe that it is possible that co-expression of IFIT1 with other IFITs from a given species might potentially enhance antiviral activity. Future studies may shed light on this.

(2) Figure 5 - Calculating the Kd for each protein would be more informative. How does the binding affinity of these IFIT1 proteins compare to that which has previously been reported?

We are unable to accurately determine kD as there is not substantial diminished signal of the free probe. Therefore, we are only able to compare IFIT1 protein binding between species without accurate mathematical calculation of binding affinity. Our result does appear similar to that of mouse Ifit1 binding to VEEV RNA (PMID: 24482115), in which the authors also do not calculate a kD for their RNA EMSA.

(3) Mutants 364 and 366 may not have direct contact with RNA, but RNA EMSA data presented suggest that the binding affinity may be different (though this is hard to conclude without Kd data). Additional biochemical data with these mutants might provide more insight here.

We agree that further studies using 364 and 366 double mutant human and chimpanzee protein in EMSAs would provide additional biochemical data and provide insight into the role of these residues in direct RNA binding. We acknowledge this is a limitation of our study as we provide only genetic data demonstrating the importance of these residues.

(4) Given that there appears to be very little overlap observed in orthologs that inhibited the viruses tested, it's possible that other amino acids may be key drivers of antiviral activity in these other orthologs. Thus, it's difficult to conclude whether the findings that residues 362/4/6 are important for IFIT1 activity can be broadly applied to other orthologs. A more systematic assessment of the role of these mutations across multiple diverse orthologs would provide more insight here. Do other antiviral proteins show this trend (ie exhibit little overlap in orthologs that inhibit these viruses). What do you think might be driving this?

We agree that other residues outside of 364 and 366 may be key drivers of antiviral activity across the IFTI1 orthologs tested. We do not hypothesize that this will broadly apply across IFIT1 from diverse clades of mammals as overall amino acid identity can differ by over 30%. However, based on the chimpanzee and human IFIT1 data, as well as sequence alignment within primates specifically, we believe these residues may be key for primate (but not necessarily other clades of mammals) IFIT1 antiviral activity.

Regarding if other antiviral proteins show little overlap in orthologs that inhibit a given virus, to our knowledge such a functional study with this large and divergent dataset of orthologs has not been performed. However, there are many examples of restriction factors exhibiting speciesspecific antiviral activity when ortholog screens have been performed. For example, HIV was reported to be suppressed by MX2 orthologs from human, rhesus macaque, and African green monkey, but not sheep or dog MX2 (PMID: 24760893). In addition, foamy virus was inhibited by the human and rhesus macaque orthologs of PHF11, but not the mouse and feline orthologs (PMID: 32678836). Furthermore, studies from our lab have shown variability in RTP4 ortholog antiviral activity inhibition towards viruses much as hepatitis C virus (HCV), West Nile virus (WNV), and Zika virus (ZIKV) (PMID: 33113352).